# Zonotope Domains for Lagrangian Neural Network Verification

**Matt Jordan**[*]
UT Austin
mjordan@cs.utexas.edu

**Jonathan Hayase**[*]
University of Washington
jhayase@cs.washington.edu

**Alexandros G. Dimakis**
UT Austin
dimakis@austin.utexas.edu

**Sewoong Oh**
University of Washington
sewoong@cs.washington.edu

## Abstract

Neural network verification aims to provide provable bounds for the output of a neural network for a given input range. Notable prior works in this domain have either generated bounds using abstract domains, which preserve some dependency between intermediate neurons in the network; or framed verification as an optimization problem and solved a relaxation using Lagrangian methods. A key drawback of the latter technique is that each neuron is treated independently, thereby ignoring important neuron interactions. We provide an approach that merges these two threads and uses zonotopes within a Lagrangian decomposition. Crucially, we can decompose the problem of verifying a deep neural network into the verification of many 2-layer neural networks. While each of these problems is provably hard, we provide efficient relaxation methods that are amenable to efficient dual ascent procedures. Our technique yields bounds that improve upon both linear programming and Lagrangian-based verification techniques in both time and bound tightness.

## 1 Introduction

With the growing prevalence of machine learning in real-world applications, the brittleness of deep learning systems poses an even greater threat. It is well-known that deep neural networks are vulnerable to adversarial examples, where a minor change in the input to a network can cause a major change in the output [1]. There is a long history of defense techniques being proposed to improve the robustness of a network, only to be completely broken shortly thereafter [2]. This has inspired researchers to focus instead on *neural network verification*, which, for a given network and a range of input, aims to verify whether a certain property is satisfied for every input in that range. A typical adversarial attack seeks to minimize the output of a scalar-valued network subject to certain input constraints. Verification provides lower bounds on the minimum output value achievable by such an adversary. Concretely, for a scalar-valued network $f$ and input range $\mathcal{X}$, verification provides lower bounds to the problem $\min_{x \in \mathcal{X}} f(x)$.

As observed in [3], prior works in verification can be broadly categorized into primal and dual views. In the primal view, convex relaxations are applied to attain, for every intermediate layer of the network, a convex superset of the true attainable range. For example, if $f = f_L \circ \cdots \circ f_1$, convex sets $\mathcal{Z}_k$ are obtained such that $\{f_k \circ \cdots \circ f_1(x) \mid x \in \mathcal{X}\} \subseteq \mathcal{Z}_k$ for $k \in \{1, \ldots, L\}$.

---

[*]Equal contribution
[†]Github Repo: https://github.com/revbucket/dual-verification

36th Conference on Neural Information Processing Systems (NeurIPS 2022).

Under the dual lens, verification is treated as a stagewise optimization problem and Lagrangian relaxation is applied to yield a dual function that always provides valid lower bounds. Often, this dual function is decomposable into a sum of minimization problems which are efficiently computable. One hallmark of all existing dual verification techniques is that intermediate bounds $\mathcal{Z}_k$ are required. In this case, either an efficient primal verification algorithm must first be applied or, for example, the dual verifier can be run iteratively on each neuron to provide upper and lower bounds for each intermediate layer.

Our approach is an attempt to combine the primal and dual threads of verification. We first apply a primal verification algorithm that generates bounds on the attainable range of each intermediate layer. To do this we leverage zonotopes, a more expressive class of polytopes than axis-aligned hyperboxes. Notably, we offer an improvement to existing zonotope bounds when we are also provided with incomparable hyperbox bounds. These zonotopic intermediate bounds are then applied to a dual verification framework, for which dual ascent can be performed. Our formulation may be viewed as taking the original nonconvex verification problem and decomposing it into many subproblems, where each subproblem is a verification problem for a 2-layer neural network. However, as even verifying a 2-layer network is hard in general, we further develop efficient relaxation techniques that allow for tractable dual ascent.

We introduce a novel algorithm, which we call ZonoDual, which *i*) is highly scalable and amenable to GPU acceleration, *ii*) provides tighter bounds than both the prior primal and dual techniques upon which our approach is built, *iii*) is highly tunable and able to effectively balance the competing objectives of bound tightness and computation speed, and *iv*) is applicable as an add-on to existing dual verification frameworks to further boost their performance. We apply ZonoDual to a variety of networks trained on MNIST and CIFAR-10 and demonstrate that ZonoDual outperforms the linear programming relaxation in both tightness and runtime, and yields a tighter bounding algorithm than the prior dual approaches.

We first discuss prior works in the verification domain. Then, we examine the existing dual framework that serves as the backbone for our algorithm, paying particular attention to the areas in which this may be tightened. Then, we discuss how we attain zonotope-based intermediate bounds and introduce the fundamental sub-problem required by our dual ascent algorithm. Ultimately, we combine these components into our final algorithm and demonstrate the scalability and tightness of our approach on networks trained on the MNIST and CIFAR-10 datasets.

## 2 Related Work

Neural network verification is a well-studied problem and has been approached from several angles. Exactly solving the verification problem is known as *complete verification*, and is known to be NP-hard even for 2-layer ReLU networks [4]. Complete verification approaches include mixed-integer programs, satisfiability modulo theories (SMT), geometric procedures, and branch-and-bound techniques [5, 6, 7, 4, 8, 9, 10]. In particular, branch-and-bound procedures generate custom branching rules to decompose verification into many smaller subproblems, which can be relaxed or further branched, allowing the algorithm to focus its efforts on the areas for greatest improvement [11, 12]. We stress that this current work focuses only on the 'bound' phase, and our approach may be applied to branched subproblems.

Often it is not necessary to exactly solve the verification problem, and only providing a lower bound to the minimization problem can suffice. This is known as *incomplete verification*. We closely examine the primal and dual verification threads, but first mention a few lines of work that do not fit neatly into this paradigm: notable approaches here include semidefinite programming relaxations [13, 14], or verification via provable upper bounds of the Lipschitz constant [15, 16, 17, 18].

**Primal verification:** A general theme in primal verification techniques is to generate convex sets that bound the attainable range of each intermediate layer. The simplest such approach is interval bound propagation (IBP) which computes a bounding hyperbox for each layer [19]. A more complex approach is to bound each intermediate layer with a polytope. Several different polyhedral relaxations are available, with DeepPoly being on the lesser-complexity side, PLANET being the canonical triangle relaxation, and the formulation by Anderson et. al being tightest albeit with exponential complexity [20, 21, 7]. Zonotopes provide a happy medium, being both highly expressive and computationally easy to work with [22, 23]. We will discuss zonotope relaxations more thoroughly in section 4.

**Dual verification:** Orthogonal to the primal approaches exist several works which consider verification by relaxations in a dual space [24, 25, 12, 26, 27]. Here, verification is viewed as an optimization problem with constraints enforced by layers of the network. The general scheme here is to introduce dual variables penalizing constraint violation, and then leverage weak duality to provide valid lower bounds. The standard Lagrangian relaxation was first proposed in [24], and then several improvements have been made to provide provably tighter bounds, faster convergence, or generalizations of the augmented lagrangian. These techniques are highly scalable and notably, one recent work has been able to provide bounds tighter than even the standard LP relaxation [9]. Our approach will follow a similar scheme, but differs fundamentally from prior works in that we operate over a tighter primal domain and can therefore provide tighter bounds.

## 3 Lagrangian Decomposition

**Problem Definition:** In a canonical neural network verification problem, we are given a *scalar-valued* feedforward neural network, $f(\cdot)$, composed of $L$ alternating compositions of affine layers and elementwise ReLU layers. We name the intermediate representations as $z_0 \ldots z_L$, and are supplied with a subset, $\mathcal{X}$, of the domain of $f(\cdot)$, which we assume to be an axis-aligned hyperbox. Verification is then written as an optimization problem

$$\underset{x_0 \in \mathcal{X}, z_0, \ldots, z_L}{\text{minimize}} \quad z_L \tag{1a}$$

$$\text{subject to} \quad z_0 = W_0 x_0 \tag{1b}$$

$$z_{k+1} = W_{k+1}\sigma(z_k) \quad (0 \le k \le L-1) \tag{1c}$$

where $\sigma$ denotes the ReLU operator applied elementwise, and each $W_k$ is the weight of an affine or convolutional layer. Our formulation is also amenable to bias terms everywhere, though we omit these throughout for simplicity.

**Dual verification:** While there are several verification approaches leveraging Lagrangian duality, we consider the Lagrangian decomposition formulation introduced in [25]. Here, each primal variable $z_k$ of the verification problem is replaced by a pair of primal variables $z_k^A, z_k^B$, along with accompanying equality constraints, yielding an equivalent optimization problem:

$$\underset{x_0 \in \mathcal{X}, z_0^A, \ldots, z_{L-1}^B, z_L^A}{\text{minimize}} \quad z_L^A \tag{2a}$$

$$\text{subject to} \quad z_0^A = W_0 x_0 \tag{2b}$$

$$z_{k+1}^A = W_{k+1}\sigma(z_k^B) \quad (0 \le k \le L-1) \tag{2c}$$

$$z_k^A = z_k^B \quad (0 \le k \le L-1) \tag{2d}$$

By introducing unconstrained dual variables $\rho_k$ for each constraint in Equation (2d) and maximizing over $\rho_k$, we attain an optimization problem that is equivalent to the original. To tractably solve this, two relaxations are performed. First, weak duality is applied by swapping the max and min, which introduces the dual function $g(\rho)$, for which every value of $\rho$ provides a lower bound:

$$g(\rho) \coloneqq \underset{x_0 \in \mathcal{X}, z_0^A, \ldots, z_{L-1}^B, z_L^A}{\text{minimize}} \quad z_L^A + \sum_{k=0}^{L-1} \rho_k^\top (z_k^A - z_k^B) \tag{3a}$$

$$\text{subject to} \quad z_0^A = W_0 x_0 \tag{3b}$$

$$z_{k+1}^A = W_{k+1}\sigma(z_k^B) \quad (0 \le k \le L-1), \tag{3c}$$

The goal now is to solve $\max_\rho g(\rho)$. The dual function can be decomposed into $L$ subproblems by substituting equality constraints and rearranging the objective function:

$$g(\rho) \coloneqq \left( \min_{x_0 \in \mathcal{X}} \rho_0^\top W_0 x_0 \right) + \sum_{k=1}^{L-2} \left( \min_{z_k^B} \rho_{k+1}^\top W_{k+1}\sigma(z_k^B) - \rho_k^\top z_k^B \right) + \tag{4a}$$

$$\left( \min_{z_{L-1}^B} W_L \sigma(z_{L-1}^B) - \rho_{L-1}^\top z_{L-1}^B \right) \tag{4b}$$

In its current state, it is unlikely that $g(\rho)$ for nonzero $\rho$ will lead to any finite lower bound as each $z_k^B$ is unconstrained. In other words, only subproblem, (4a), has any information about the input

constraint ($x_0 \in \mathcal{X}$). To remedy this, one can tighten the dual optimization by imposing *intermediate bounds* on $z_k^B$'s, so long as those bounds include all feasible values. Specifically, for each $z_k^B$, we can impose a bounding set $\mathcal{Z}_k$, as long as the following implication holds:

$$x_0 \in \mathcal{X} \implies z_k^B \in \mathcal{Z}_k \qquad 1 \le k \le L - 1. \tag{5}$$

All prior dual approaches have chosen the sets $\mathcal{Z}_k$ to be axis-aligned hyperboxes for two reasons. First, it is extremely efficient to attain hyperbox bounds at every layer. Second, when $\mathcal{Z}_k$ is a hyperbox, the dual function can be evaluated efficiently by further decomposing each component of the dual along its coordinates. Equipped with intermediate bounds $\mathcal{Z}_k$, and noting that $g(\rho)$ is a concave function of $\rho$, the standard procedure is to perform dual ascent on $g(\rho)$. This requires gradients $\nabla_\rho g(\rho)$, which can easily be computed as the primal residuals. Letting $z_k^{A*}$ and $z_k^{B*}$ be the argmin of the dual function, then $\nabla_{\rho_k} g(\rho) = z_k^{A*} - z_k^{B*}$.

The key innovation in this work is to replace the intermediate bounds $\mathcal{Z}_k$ with zonotopes everywhere. In this case, intermediate bounds may easily be computed as we will see in the next section. Zonotopes have a distinct advantage over hyperboxes in that every coordinate is no longer independent, and therefore neuron dependencies are encoded. This further constrains the feasible set of the dual function and leads to larger dual values. However, this comes with the cost that the dual function is more computationally difficult to evaluate.

## 4 Zonotopes

We start with a review of zonotopes and how they may be used to attain intermediate bounds. Then we introduce the problem of ReLU programming, tie it to the dual decomposition formulation, and discuss our relaxations for efficient dual evaluation. Proofs of all claims are contained in the appendix.

**Zonotope Properties:** Zonotopes are a class of polytopes, which we formally define using the notation $Z(c, E)$ to refer to the set

$$Z(c, E) := \{c + Ey \mid y \in [-1, 1]^m\}$$

where $c \in \mathbb{R}^d$ is called the *center*, and $E \in \mathbb{R}^{d \times m}$ is called the *generator matrix*. Each column of the generator matrix is a line segment in $\mathbb{R}^d$, and $Z(c, E)$ can be equivalently be viewed as the Minkowski sum of each generator column offset by $c$; or as the affine map $y \to c + Ey$ applied to the $\ell_\infty$ ball in $\mathbb{R}^m$. In particular, note that a hyperbox is a zonotope with a diagonal generator matrix. Zonotopes have several convenient properties:

*Efficient Linear Programs:* Linear programs can be solved in closed form over a zonotope:

$$\min_{z \in Z(c, E)} a^\top z = a^\top c + |a^\top E| \vec{1}.$$

*Closure under affine operators and Minkowski sums:* given an affine map $x \to Wx + b$, the set $\{Wz + b \mid z \in Z(c, E)\}$ is equivalent to the zonotope $Z(Wc + b, WE)$. The Minkowski sum of two sets $A, B$ is defined as $A \oplus B := \{x + y \mid x \in A, \quad y \in B\}$. Letting $E_1 || E_2$ denotes concatenation of columns, the Minkowski sum of two zonotopes $Z(c_1, E_1)$, $Z(c_2, E_2)$ is also a zonotope:

$$Z(c_1, E_1) \oplus Z(c_2, E_2) = Z(c1 + c2, E_1 || E_2).$$

**Intermediate Bounds with Zonotopes:** Prior work has developed efficient techniques to generate intermediate layer bounds using zonotopes for feedforward neural networks. The details are deferred to the appendix, and we can assume that we have access to a black box that, given a zonotope $\mathcal{Z}$, is able to generate a zonotope $\mathcal{Y}$ such that $\{\sigma(z) \mid z \in \mathcal{Z}\} \subseteq \mathcal{Y}$. Such an operation is called a *sound pushforward operator*. One such pushforward operator is known as `DeepZ` and can be applied repeatedly to provide valid zonotopic bounds for every intermediate layer of the neural network [22]. We make two observations regarding these intermediate bounds here:

**Proposition 4.1.** *Let $\mathcal{Z}_k$ be the $k^{th}$ zonotope bound provided by the `DeepZ` procedure, and let $\mathcal{H}_k$ be the $k^{th}$ intermediate bound provided by the Kolter-Wong dual procedure [20]. Then the smallest hyperbox containing $\mathcal{Z}_k$ is exactly $\mathcal{H}_k$.*

**Proposition 4.2.** *Suppose $\mathcal{Z}$ is a zonotope and $\mathcal{H}$ is a hyperbox, such that $\mathcal{Z} \not\subseteq \mathcal{H}$. Then we can develop a sound pushforward operator for the ReLU operator that outputs a zonotope $\mathcal{Z}'$ containing the ReLU of every element of $\mathcal{Z} \cap \mathcal{H}$, with the property that $\mathcal{Z}' \subsetneq \text{DeepZ}(\mathcal{Z})$.*

The first proposition states that a common technique for generating intermediate hyperbox bounds is never any tighter than the procedure we use to generate intermediate zonotope bounds. The second proposition describes an improved pushforward operator that offers improvements against `DeepZ` but requires extra information provided by a hyperbox.

**ReLU Programming:** We return our attention to the dual function introduced in equations Equations (4a) and (4b). Note that the first term of Equation 4a is a linear program of a zonotope which can be solved extremely efficiently, while all other terms adopt a form we refer to as a *ReLU program*

**Definition 4.1.** *Given a set $\mathcal{Z} \subseteq \mathbb{R}^d$, and two objective vectors $c_1, c_2 \in \mathbb{R}^d$, we say the **ReLU Program** is the optimization problem*

$$\min_{z \in \mathcal{Z}} c_1^\top z + c_2^\top \sigma(z). \tag{6}$$

ReLU programming is nonconvex for general $c_2$. When $\mathcal{Z}$ is an interval, only the endpoints, or 0 if it is contained in the interval, are candidate optima. In this case, the ReLU program may be solved by evaluating the objective at each candidate. When $\mathcal{Z}$ is a hyperbox in $\mathbb{R}^d$, each coordinate can be considered independently, amounting to solving $d$ ReLU programs over intervals. The story quickly becomes more complicated if $\mathcal{Z}$ is a zonotope:

**Theorem 4.1.** *When $\mathcal{Z}$ is a zonotope, solving a ReLU program over $\mathcal{Z}$ is equivalent to a neural network verification problem for a 2-layer network, which is known to be NP-hard.*

**Remarks:** This theorem implies that solving ReLU programs over more complex sets, such as polytopes in general, is also NP-hard. However it remains an open question whether solving ReLU programs over other simpler sets such as parallelepipeds, i.e. zonotopes with an invertible generator matrix, is hard. Note that ReLU programs can be solved exactly via a Mixed-Integer Program (MIP) using the standard form for encoding a ReLU [5].

We pause here to observe the current situation. Lagrangian methods for verification first make a relaxation by applying weak duality. When using zonotopes for the intermediate bounds, one is able to decompose the very large non-convex optimization problem of deep network verification into many smaller subproblems which are equivalent to two-layer network verification problems. Unfortunately, each of these subproblems is also NP-hard, and thus must be further relaxed to attain tractability.

**Zonotope Partitioning:** The primary relaxation technique we consider comes from the following observation. Consider a $d$-dimensional zonotope $\mathcal{Z}$ and ReLU programming objectives $(c_1, c_2)$. Then, for any partition $S_1 \cup \cdots \cup S_n$ of $\{1, \ldots, d\}$, the inequality

$$\min_{z \in \mathcal{Z}} \sum_{i=1}^d \left( c_{1,i} z_i + c_{2,i} \sigma(z_i) \right) \geq \sum_{j=1}^n \min_{z \in \mathcal{Z}} \sum_{i \in S_j} \left( c_{1,i} z_i + c_{2,i} \sigma(z_i) \right)$$

holds. This approach has a nice geometric interpretation as well. The decomposed minimization $\min_{z \in \mathcal{Z}} \sum_{i \in S_j} \left( c_{1,i} z_i + c_{2,i} \sigma(z_i) \right)$ can be viewed as optimization over $\mathcal{Z}$ when it has been projected onto the linear subspace spanned by the elements of $S_j$. Projection onto a linear subspace is an affine operator, so each projection of $\mathcal{Z}$ is itself a zonotope. Indeed, the projection onto the linear subspace spanned by $S_j$ is simply the center and generator rows indexed by $S_j$.

Observe that if each $S_j$ were a singleton set, then decomposing the ReLU program according to these $S_j$'s is equivalent to relaxing the ReLU program by casting $\mathcal{Z}$ to the smallest containing hyperbox. If the partition were the trivial partition $\{1, \ldots, d\}$, then $\mathcal{Z}$ remains unchanged. Finally, notice that the Minkowski sum of every projection of $\mathcal{Z}$ is itself a zonotope.

In this sense, by choosing any partitioning of coordinates, we are able to relax $\mathcal{Z}$ into a zonotope that is simultaneously a superset of $\mathcal{Z}$ and a subset of the hyperbox-relaxation of $\mathcal{Z}$. This is optimistic for our dual ascent procedure, as any partitioning restricts the primal feasible space for the dual function evaluation. It is in this way that we are able to interpolate the coarseness of our relaxation, which is directly controlled by the coarseness of our partitioning. Because we expect the complexity of solving a ReLU program over a zonotope to be superlinear in the dimension, shattering a zonotope in this fashion can drastically improve the complexity of solving a relaxation of a ReLU program.

**Two-dimensional Zonotopes:** A special case we consider is when we partition a zonotope into 2-dimensional groups. This is because a 2-dimensional zonotope can be significantly more descriptive

than a 2-dimensional rectangle, but the solution of a ReLU program can still be computed efficiently. Observe that the argmin of a ReLU program over a 2-dimensional zonotope must occur at either i) a vertex of the zonotope, ii) a point where the zonotope crosses a coordinate-axis, iii) the origin. This follows from the fact that a ReLU program over 2-dimensions may be viewed as 4 independent linear programs over the intersection of the zonotope and each orthant. For the purposes of ReLU programs, it suffices to enumerate all vertices of the zonotope and compute any axis crossings or origin-containments. It turns out that this may be done in nearly linear time:

**Theorem 4.2.** *If $Z(c, E)$ is a 2-dimensional zonotope with $m$ generators, $Z$ has $2m$ vertices and the set of all vertices, axis crossings, and the containment of the origin can all be computed in $O(m \log m)$ time.*

## 5   The ZonoDual Algorithm

With both preliminaries of dual verification techniques and zonotopic primal verification techniques in hand, we can now fully describe the algorithm we employ in practice. We first describe the algorithm as it is used to bound a single neuron's value, i.e. the output neuron. After we describe how we adapt our approach to the stagewise setting where our algorithm is applied to each intermediate neuron to yield tighter box bounds. Our algorithm can be described in three main phases: an *initialization phase*, where primal bounds are generated to define the feasible set for each subproblem in the dual function; an *iteration phase*, where the argmin of the dual function is computed many times to provide informative gradients for updates to the dual variables; and an *evaluation phase*, where the primal bounds are tightened to yield a more computationally intensive dual function, which only needs to be evaluated a single time.

**Initialization phase:** Naively, we can apply the `DeepZ` procedure here to attain a zonotope bounding the output range of the network at every layer. Both this procedure and our box-improved variant can be done extremely efficiently. We adapt two techniques from [25] in the initialization phase. First we notice that some very minor improvements to `DeepZ` can be made using an application of Proposition 4.2 when provided with the boxes from interval bound propagation. Second, we initialize the dual variables to the KW dual variables from [20], which by 4.1 are equivalent to the scale factors applied internally in the zonotope pushforward operators.

**Iteration Phase:** In the iteration phase, each intermediate zonotope $\mathcal{Z}_k$ is partitioned into components that are amenable to repeated evaluations of the dual function. We always initially partition each zonotope into 2-d chunks for which the dual function may be evaluated on a GPU without requiring any MIP calls. Recall that for a $d$-dimensional zonotope with $m$ generators, after the initial $O(dm \log m)$ candidate optima computation, each ReLU program can be evaluated in $O(dm)$ time. These initial 2-d partitions are computed using one of several simple heuristics, described in the appendix, with the aim to generate 2-d zonotopes which are as 'un-box-like' as possible. Optionally, after an initial 2-d iteration phase, these partitions may be combined and dual ascent can be applied, where each subproblem now requires several calls to a MIP solver. At any point during this phase, the dual function evaluations still provide valid lower bounds to the verification problem.

**Evaluation Phase:** In this phase, we can presume that effective dual variables have been found during the previous phase. The objective here is to tighten the feasible range of each subproblem, which will necessarily improve the value of the dual function. Because we only need to evaluate a bound on the dual function once here, we find that it is often worth it to spend more time solving each subproblem. We do this by simply merging several partition elements together, converting our 2-dimensional zonotopes into higher-dimensional ones that require calls to a MIP solver to compute. Despite the theoretical intractability of MIPs, we find that this procedure is remarkably efficient. In practice, it is often sufficient to only merge zonotopes at the later layers in a network where the dual function subproblems are the most negative. Finally, we note that as only a *bound* upon the dual function is required here, and each subproblem is itself a 2-layer verification problem, the MIPs may be terminated early. Any other incomplete verification approach can be applied here as well.

**Guarantees:** Ultimately, our algorithm will always provide a valid lower bound to the verification problem so long as the dual function is evaluated appropriately. The provable correctness of our algorithm stems from i) the fact that weak duality always holds and any evaluation of the dual function with feasible dual variables provides a valid lower bound tot the primal problem; and ii) by leveraging zonotopes and a sound relaxation of them, we only ever provide lower bounds to the evaluation of the dual function. Further, we can also guarantee that, for any fixed set of dual variables $\rho$, the bounds

we provide will be tighter than those provided by the approach in [25]. This follows because we have reduced the feasible set within each dual subproblem, relative to the hyperbox approaches in prior works. Similarly, because $\rho = 0$ is always a valid choice, when the dual variables are optimized appropriately, we will provide tighter bounds than whichever primal verification method is applied at the initialization phase.

**Stagewise computation:** When the network of interest is of sufficiently small size, any verification algorithm may be applied stagewise, where the algorithm computes upper and lower bounds to each neuron, starting from those in the first layer. This is often effective for generating tighter intermediate box bounds, which in turn yields much tighter output bounds. However this comes the cost of running many calls to the verification algorithm. This can be parallelized for verification procedures amenable to GPU acceleration, but demands a large memory footprint.

At first glance, it may seem that such a stagewise procedure obviates the improvements we attain by leveraging zonotopes: indeed, each stagewise bound is a *box bound*. However, we show that we can apply this stagewise trick to our approach to generate tighter box bounds, which can then be employed by Proposition 4.2 to yield tighter zonotopic bounds. Additionally, in the case that these box bounds are incomparable to the improved zonotope bounds, these can be employed in imposing further constraints upon each subproblem. Instead of solving each ReLU program over a zonotope, we can instead solve each ReLU program over the intersection of a zonotope and a hyperbox. In the higher-dimensional case, it is trivial to apply elementwise constraints to each variable in the MIP — in fact, this often improves the running time of each MIP. In the 2-d case, we observe that we are able to solve ReLU programs over the intersection of a 2-d zonotope and rectangle as efficiently as solving over a 2-d zonotope alone. This is the content of the following corollary to Theorem 4.2.

**Corollary 5.1.** *Given a 2-d zonotope with $m$ generators, the set of candidate optimal points to a ReLU program has cardinality no greater than $(2m + 9)$ and is computable in time $O(m \log m)$.*

**Ablation study:** To clarify the bound improvements and their corresponding runtime cost imposed by each step of our approach, we perform an ablation study on an MNIST fully connected network (MNIST FFNet). We report the bounds relative to the bound attained by the LP relaxation in Table **??**. We consider three initialization settings separately: the standard `DeepZ` zonotopes, the `DeepZ` zonotopes augmented with IBP bounds, and zonotopes informed by the the box bounds from BDD+ [25]. Then we perform dual ascent over 2-d zonotopes, before merging the zonotopes of the final layer only and evaluating the MIP. We report these numbers in Table **??**. Here we see that a tight initialization does indeed help, but this gap shrinks after our dual ascent procedure. We also see a substantial gain in bound improvement by using MIPs, even in the last layer's zonotopes alone.

## 6 Experiments

We evaluate the effectiveness of our approach on several networks trained on the MNIST and CIFAR-10 datasets. While many recent verification techniques employ a branch-and-bound strategy, we only provide innovations on the bounding component of this pipeline. Strong verifiers should employ both effective bounding algorithms and clever branching strategies, which are both necessary but fairly independent components. Like other dual-based verification techniques, our approach can be applied using any of a number of branching strategies. We focus our efforts primarily on the improvements in the *bound* provided by a single run of each considered method. We argue this is a reasonable choice of metric because most prior benchmarks focus on $\epsilon$-values far lower than typically used in adversarial attack scenarios. This artifically decreases the depth to which branching strategies must search, skewing the results. By focusing on the bound of the verification, we directly measure the metric that bounding algorithms such as ours seek to optimize. However, for the sake of completeness, we do also evaluate verified robustness percentage upon several benchmark MNIST networks. For bound evaluations, we separate our experiments into two cases: we first examine the setting where only the output neuron is optimized, which we denote the 'single-stage' setting; and the case where every neuron is optimized to provide tighter intermediate box bounds, denoted the 'stagewise' setting. We compare the bound and running time against the following approaches: **BDD+**, the Lagrangian Decomposition procedure proposed in [25], where the optimization is performed using optimal proximal step sizes; **LP**, the original linear programming relaxation using the PLANET relaxation on each ReLU neuron [20]; **Anderson**, the linear programming relaxation, where 1-cut is applied per neuron as in [7]; **AS**, a recent Lagrangian technique which iteratively chooses cuts to employ from the Anderson relaxation [9].

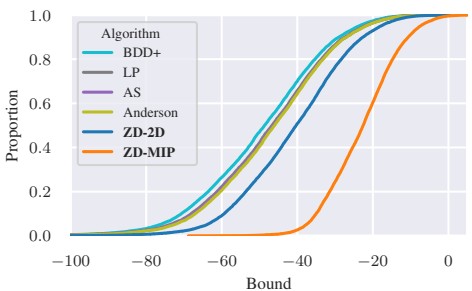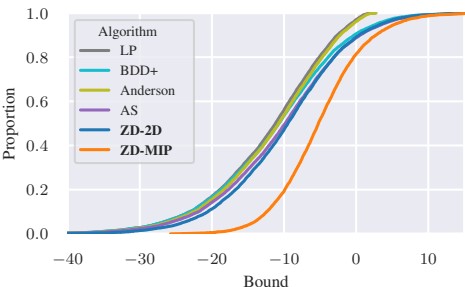

Figure 1: Cumulative density function plots of the bounds obtained by each algorithm for a deep convolutional network trained on MNIST (left) and a deep convolutional network trained on CIFAR10 (right). Lines further to the right indicate tighter bounds. Our algorithms are bolded in the legend.

All baselines are run implemented using the code and hyperparameter choices from prior works, and network weights are taken from existing works where applicable. Architecture descriptions are contained in the supplementary. We follow [25] to construct scalar-valued verification problems: for examples with label $i$, we compare the difference between the $i^{th}$ and $(i+1)^{th}$ logit of a network. We can extend this procedure to a multi-class task by applying the verification $n-1$ times for each incorrect logit.

Table 1: Timing data for the single-stage bounds.

| Alg\Net | MNIST Deep | CIFAR SGD |
|---|---|---|
| BDD+ | $2.5 \pm 0.2$ | $1.5 \pm 0.1$ |
| LP | $4.8 \pm 0.5$ | $6.2 \pm 0.6$ |
| **ZD-2D** | $10.9 \pm 0.5$ | $7.6 \pm 0.3$ |
| AS | $16.9 \pm 1.2$ | $10.2 \pm 0.8$ |
| **ZD-MIP** | $23.1 \pm 12.4$ | $11.0 \pm 3.1$ |
| Anderson | $52.4 \pm 27.5$ | $21.3 \pm 4.6$ |

**Single-stage bound evaluation:** In the single-stage setting we make the following hyperparameter choices for our approach. We initialize the intermediate bounds using zonotopes informed by the boxes attained by the best of the `DeepZ` bounds and the IBP bounds. We then decompose each zonotope into 2-dimensional partitions, using the 'similarity' heuristic for fully-connected networks, and the 'spatial' heuristic for convolutional networks. Then we perform 1000 iterations of the Adam optimizer during the dual ascent procedure. We report the value at the last iteration as **ZD-2D**. Then we merge the partitions of only the last layer of each network into 20-dimensional zonotopes, and evaluate the dual function under this new partitioning. We report these values as **ZD-MIP**.

We present results of our approach on the MNIST-Deep network, which has 5196 neurons and 6 layers, evaluating over a domain of $\ell_\infty$ boxes with $\epsilon = 0.1$. Specifically we present cumulative density function plots: a point $(x, y)$ on any curve in these plots indicates that that particular algorithm attains a bound tighter than $x$ for $(1-y)\%$ of the examples. Curves further to the right indicate tighter bounds in aggregate. Figure 1 (left) contains the distribution of reported bounds over every correctly-classified example in the MNIST validation set. In Figure 1 (right), we present a similar CDF plot for a CIFAR-10 network with 6244 ReLU neurons and 4 layers. For this CIFAR network, we set $\epsilon = 5/255$, as in [9]. The average and standard deviation runtimes for these networks is provided in Table 1.

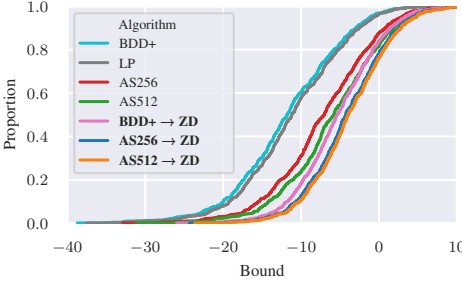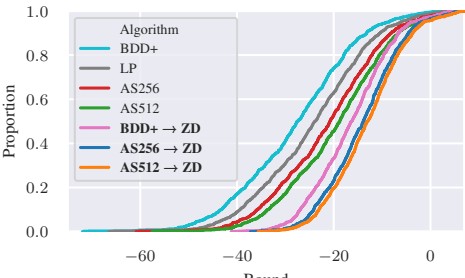

Figure 2: Stagewise bounds on the MNIST-FFNet (left) and MNIST-Deep (right) networks.

**Stagewise bound evaluation:** Now we turn our attention to the setting where enough computation time is allowed to perform the optimization procedure on every neuron in turn, which is able to

Table 3: Verified Robustness on MNIST Networks. Methods denoted with an asterisk had numbers taken from [28].

| Model | $\epsilon$ | **ZD** | BDD+ | CROWN* | PRIMA* | $\beta$-CROWN* |
|---|---|---|---|---|---|---|
| MLP 5x100 | 0.026 | 28.5% (24s) | 19.8% (7s) | 16.0% (1s) | 51.0% (159s) | 69.9% (102s) |
| MLP 8x100 | 0.026 | 21.8% (19s) | 19.1% (15s) | 18.2% (1s) | 42.8% (301s) | 62.0% (103s) |
| MLP 5x200 | 0.015 | 41.8% (28s) | 33.2% (8s) | 29.2% (2s) | 69.0% (224s) | 77.4% (86s) |
| MLP 8x200 | 0.015 | 29.7% (26s) | 27.4% (17s) | 25.9% (6s) | 62.4% (395s) | 73.5% (95s) |
| ConvSmall | 0.12 | 52.6% (38s) | 16.6% (6s) | 15.8% (3s) | 59.8% (42s) | 72.7% (7s) |

generate tighter box bounds. Here we notice an interesting observation: our method can be improved by leveraging tighter intermediate box bounds, but is agnostic as to where these bounds come from. Indeed, our approach can take as input any collection of tighter box bounds and generate tighter zonotopes via Proposition 4.2. However, our current implementation of this tighter primal bound is not vectorized to be amenable, in the way that others are, to stagewise bounding procedures. This is purely an implementation detail, and we instead use existing primal bounding techniques.

As baselines here, we compare against the LP approach, BDD+, and the active set approach with 256 and 512 iterations, denoted AS256, AS512. We leverage box bounds provided by each of these dual approaches to inform our zonotope bounds from which we can run our method, denoted by a $\to$ ZD. Note that BDD+ can never surpass bounds provided by the LP approach, whereas ours and the active set procedure can both improved even further with a longer runtime. Hence it is not sufficient to provide a better bound, but to also do so more efficiently. In this case, our approach performs 2000 iterations of Adam on 2-D zonotopes and increases the partition size of the final two layers of each network to sizes of 16 and 20 respectively.

We present our results on the MNIST-FFNet and MNIST-Deep network which have 960 and 5196 neurons, and 5 and 6 layers respectively, again using an $\epsilon = 0.1$. CDF plots of bounds for these two networks are provided in 2, with timing data in 2. We observe that the BDD+ bound completes very quickly, but attains a bound only slightly worse than LP. Our approach, when initialized with the BDD+ bounds runs more efficiently than AS256, providing bounds that are comparable or better than the slower AS512 approach.

Table 2: Timing data for stagewise bounds

| Alg\Net | MNIST Deep | MNIST Wide |
|---|---|---|
| BDD+ | $15.4 \pm 0.4$ | $8.3 \pm 0.3$ |
| **BDD+** $\to$ **ZD** | $51.7 \pm 14.0$ | $36.4 \pm 6.8$ |
| AS256 | $138.9 \pm 0.9$ | $45.4 \pm 0.6$ |
| **AS256** $\to$ **ZD** | $174.1 \pm 13.6$ | $73.3 \pm 6.6$ |
| AS512 | $289.3 \pm 1.7$ | $91.3 \pm 0.9$ |
| **AS512** $\to$ **ZD** | $324.1 \pm 13.5$ | $119.2 \pm 6.9$ |
| LP | $2132.2 \pm 494.4$ | $387.1 \pm 65.5$ |

**%-Verified Robustness:** Finally we evaluate our approach comparing against all incorrect logits on MNIST trained networks, mirroring the setup of [28]. We report numbers from an unbranched version of ZD and BDD+ alongside several other verification techniques [29, 30, 31]. Compared to other efficient, branch-friendly procedures such as BDD+ and CROWN, our approach provides tighter bounds at a fair cost to efficiency. Our approach provides weaker bounds than PRIMA, but is significantly faster. $\beta$-CROWN, in particular, leverages a dual-based branching strategy which could be applied to our method. However due to its extremely efficient bounding procedure and the need to solve millions of subproblems for verification, it is unclear if our approach can surpass this method.

## 7 Conclusion

In this work, we demonstrated how primal and dual verification techniques can be combined to yield efficient bounds that are tighter than either of them alone. Primal methods capture complex neuron dependencies, but do not optimize for a verification instance; while dual methods are highly scalable, but rely on unnecessarily loose primal bounds. This work is a first step in combining these two approaches. We believe that this combination of approaches is a necessary direction for providing scalable and tight robustness guarantees for safety-critical applications of machine learning.

**Acknowledgements:** This work is supported in part by NSF grants CNS-2002664, DMS-2134012, CCF-2019844 as a part of NSF Institute for Foundations of Machine Learning (IFML), CNS-2112471 as a part of NSF AI Institute for Future Edge Networks and Distributed Intelligence (AI-EDGE), CCF-1763702, AF-1901292, CNS-2148141, Tripods CCF-1934932, and research gifts by Western

Digital, WNCG IAP, UT Austin Machine Learning Lab (MLL), Cisco and the Archie Straiton Endowed Faculty Fellowship.

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
