| AS256 | $138.9 \pm 0.9$ | $45.4 \pm 0.6$ |
| **AS256 $\rightarrow$ ZD** | $174.1 \pm 13.6$ | $73.3 \pm 6.6$ |
| AS512 | $289.3 \pm 1.7$ | $91.3 \pm 0.9$ |
| **AS512 $\rightarrow$ ZD** | $324.1 \pm 13.5$ | $119.2 \pm 6.9$ |
| LP | $2132.2 \pm 494.4$ | $387.1 \pm 65.5$ |

**%-Verified Robustness:** Finally we evaluate our approach comparing against all incorrect logits on MNIST trained networks, mirroring the setup of [28]. We report numbers from an unbranched version of ZD and BDD+ alongside several other verification techniques [29, 30, 31]. Compared to other efficient, branch-friendly procedures such as BDD+ and CROWN, our approach provides tighter bounds at a fair cost to efficiency. Our approach provides weaker bounds than PRIMA, but is significantly faster. $\beta$-CROWN, in particular, leverages a dual-based branching strategy which could be applied to our method. However due to its extremely efficient bounding procedure and the need to solve millions of subproblems for verification, it is unclear if our approach can surpass this method.

## 7 Conclusion

In this work, we demonstrated how primal and dual verification techniques can be combined to yield efficient bounds that are tighter than either of them alone. Primal methods capture complex neuron dependencies, but do not optimize for a verification instance; while dual methods are highly scalable, but rely on unnecessarily loose primal bounds. This work is a first step in combining these two approaches. We believe that this combination of approaches is a necessary direction for providing scalable and tight robustness guarantees for safety-critical applications of machine learning.

**Acknowledgements:** This work is supported in part by NSF grants CNS-2002664, DMS-2134012, CCF-2019844 as a part of NSF Institute for Foundations of Machine Learning (IFML), CNS-2112471 as a part of NSF AI Institute for Future Edge Networks and Distributed Intelligence (AI-EDGE), CCF-1763702, AF-1901292, CNS-2148141, Tripods CCF-1934932, and research gifts by Western

Digital, WNCG IAP, UT Austin Machine Learning Lab (MLL), Cisco and the Archie Straiton Endowed Faculty Fellowship.

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

## A  Broader Impacts

Our work focuses on proving robustness and security for machine learning applications. While any new technology may be used maliciously, we believe the technical advances proposed in this work are as benign as possible and will only serve to offer certification of existing technologies.

## B  Limitations and Future Work

While ZonoDual has some benefits over prior works, it is not without limitations. We enumerate these here, as well as potential ways to overcome these:

- **Reliance on ReLU nonlinearities:** ZonoDual relies on the ability to solve ReLU programs over zonotopes using MIPs. Unfortunately, this approach cannot generalize to nonlinearites that are not piecewise linear, such as the sigmoid activation. While it might be possible to solve 'sigmoid programs' over 2-dimensional zonotopes, the formulation becomes significantly more complicated.

- **MIPs are not conducive to GPU acceleration:** A crucial component of our procedure is to evaluate the dual using a tighter relaxation during the final phase of ZonoDual. Prior works that leverage box bounds do not include such a tightening step, but we note that MIPs can not benefit from GPU computations. In practice, small MIPs can solve very quickly, but for extremely large nets, such as SOTA image models, it is unlikely that this approach can scale as nicely as other techniques. This presents a need for fast and tight lower bounds for the reduced verification problem when the network is known to only have one hidden layer. Unfortunately, such methods, such as SDP's cannot currently scale to the case where this would be helpful for us.

- **Intermediate zonotopes may have very large order:** One drawback of using zonotopic intermediate bounds is that the number of generators can grow to be quite large for deeper layers in the network. This can slow down the dual computation procedure, even in the 2-dimensional case for which dual evaluation time is linear in the number of generators. There is room for improvement here in what is known as 'order reduction' techniques for zonotopes, which soundly compresses the number of generators of a zonotope. Naive approaches are very quick and can allow for a 2x compression factor with only marginal degradations to dual function values, but significantly tighter compressions require much more work. We have not included any of these accelerations in our presentation for the sake of simplicity.

- **Exact evaluations of the dual seem to be necessary for informative gradients:** One might hope that the dual need not be evaluated exactly during the iteration phase of ZonoDual. Such a procedure could allow for much faster and tighter dual ascent iterations when only informative gradients are required. Unfortunately the approaches we have tried in this domain (including Frank-Wolfe, L-BFGBS, and a modified simplex algorithm) do not provide gradients that are informative enough to yield good dual values for the final evaluation phase. It is possible there are techniques here that could work, but we have not found them.

## C  Zonotope Pushforward Operators

Here we describe the sound zonotope pushforward operators we use to generate intermediate bounds. Since zonotopes are closed under affine transformation, all that's required is to describe the pushforward operator for the ReLU operator. Specifically, given a zonotope $Z(c, E)$, we seek to generate a zonotope $Z(c', E')$ such that $\{\sigma(z) \mid z \in Z(c, E)\} \subseteq Z(c', E')$. First we describe the approach when no extra information is known. This was first presented in [32], but we adapt the analysis of [33]. Next we consider the case when we also have the extra information in the form of a hyperbox $H(l, u) := \{x \mid l \leq x \leq u\}$, where the goal is to construct $Z(c', E')$ such that $\{\sigma(z) \mid z \in Z(c, E) \cap H(l, u)\} \subseteq Z(c', E')$.

DeepZ: The primary strategy is to compress each coordinate of $Z(c, E)$ by a scale factor $\lambda_i \in [0, 1]$, which will in general, incur some error along each coordinate, $b_i$. These errors are accumulated into a hyperbox, which will then be incorporated into $Z(c', E')$ in the form of a Minkowski sum. The goal is to minimize the amount of error that is accumulated. In other words, each coordinate can be

considered independently, where we wish to solve the minmax procedure

$$\min_{\lambda_i, b_i} \left( \max_{z \in Z(c,E)} |\sigma(z_i) - \lambda_i z_i - b_i| \right)$$

where the choice of the compression factor $\lambda_i$ is chosen to minimize the error that needs to be added in at the end between $\sigma(z_i)$ and $\lambda_i z_i$ over all feasible points in the zonotope. Noticing that projecting $Z(c, E)$ only onto the $i^{th}$ coordinate necessarily yields an interval, so the max is performed over an interval, $[l_i, u_i]$. In the case that $l_i \geq 0$, the ReLU is always on, and $\lambda_i$ can be chosen to be 1, with $b_i = 0$. In the case that $u_i \leq 0$, then the ReLU is always off, and $\lambda_i$ can be chosen to be 0, again with $b_i = 0$. In the case that $l_i < 0 < u_i$, it is not difficult to see that the optimum value is attained when $\lambda_i$ is chosen to be $\frac{u_i}{u_i - l_i}$, such that the optimal value is $\frac{-l_i u_i}{u_i - l_i}$. Hence we can accumulate $\lambda_i$ for each coordinate $i$ into a diagonal matrix $\Lambda$. The optimal offsets, $(b_1, \ldots, b_d)$, can be divided by 2, and accumulated into a diagonal matrix $B$ whose columns may be appended to the compressed zonotope. Put concisely,

$$Z(c', E') := Z(\Lambda c, \Lambda E | B)$$

where

$$\Lambda_{i,i} = \begin{cases} 0 & \text{if } u_i \leq 0 \\ 1 & \text{if } l_i > 0 \\ \frac{u_i}{u_i - l_i} & \text{otherwise} \end{cases} \qquad B_{i,i} = \begin{cases} 0 & \text{if } u_i \cdot l_i \geq 0 \\ \frac{-l_i \cdot u_i}{2(u_i - l_i)} & \text{otherwise} \end{cases}$$

**Improvements using Hyperboxes:** Now we restate our proposition that allows for an improvement upon this pushforward operator when we have an incomparable hyperbox bound.

**Proposition 4.2.** *Suppose $\mathcal{Z}$ is a zonotope and $\mathcal{H}$ is a hyperbox, such that $\mathcal{Z} \not\subseteq \mathcal{H}$. Then we can develop a sound pushforward operator for the ReLU operator that outputs a zonotope $\mathcal{Z}'$ containing the ReLU of every element of $\mathcal{Z} \cap \mathcal{H}$, with the property that $\mathcal{Z}' \subsetneq \texttt{DeepZ}(\mathcal{Z})$.*

*Proof.* Let $Z(c, E)$ be the zonotope $\mathcal{Z}$, and let $H(l^h, u^h)$ be the hyperbox $\mathcal{H}$. Similar to $\texttt{DeepZ}$, the strategy we employ to develop a sound pushforward operator is to scale each coordinate independently, and Minkwoski sum the errors, accumulated as a hyperbox. Again, this decomposes into a minmax procedure as

$$\min_{\lambda_i, b_i} \left( \max_{z \in Z(c,E) \cap H(l^h, u^h)} |\sigma(z_i) - \lambda_i z_i - b_i| \right).$$

In a single dimension, both a zonotope and hyperbox are intervals, so their intersection is also an interval. Letting $(l_i^z, u_i^z)$ be the $i^{th}$ coordinates lower and upper bounds for $\mathcal{Z}$, then the intersection between the $i^{th}$ coordinate projection of $\mathcal{H}$ and $\mathcal{Z}$ is the interval $[l_i, u_i] := [\max(\{l_i^z, l_i^h\}), \min(\{u_i^z, u_i^h\})]$. From here the choice of $\Lambda$ and $B$ is exactly the same as in $\texttt{DeepZ}$, where the tightened coordinate-wise bounds are used. This is guaranteed to be as tighter as long as at least one coordinate exists for which the interval is reduced, because both the compression factor $\lambda_i$ and error $b_i$ are reduced by tightening the interval. By our assumption that $\mathcal{Z} \not\subseteq \mathcal{H}$, at least one coordinate interval must be tightened, so the inclusion $\mathcal{Z}' \subset \texttt{DeepZ}(\mathcal{Z})$ is strict. $\square$

**Kolter-Wong Bounds** Here we present a geometric proof of Proposition 4.1. We will only briefly discuss the derivation of this bound in the context of this proof. For a more thorough discussion on the exact structure and derivation of the Kolter/Wong dual relaxation, we point to the original paper [20] and a self-contained derivation in the appendix of [25].

**Proposition 4.1.** *Let $\mathcal{Z}_k$ be the $k^{th}$ zonotope bound provided by the $\texttt{DeepZ}$ procedure, and let $\mathcal{H}_k$ be the $k^{th}$ intermediate bound provided by the Kolter-Wong dual procedure [20]. Then the smallest hyperbox containing $\mathcal{Z}_k$ is exactly $\mathcal{H}_k$.*

*Proof.* In [20], the authors derive the original linear programming relaxation and dual procedure for verifying ReLU networks. This technique can be interpreted in the primal sense as maintaining a polyhedral primal bound for every intermediate layer. Assuming a polyhedral input range, such a hyperbox, this procedure works by incrementally lifting the dimensionality of the polytope. In other words, an H-polytope is maintained, where new dimensions/variables are introduced through equality constraints defined by each layer. For affine layers, this lifting procedure is exact. For ReLU layers,

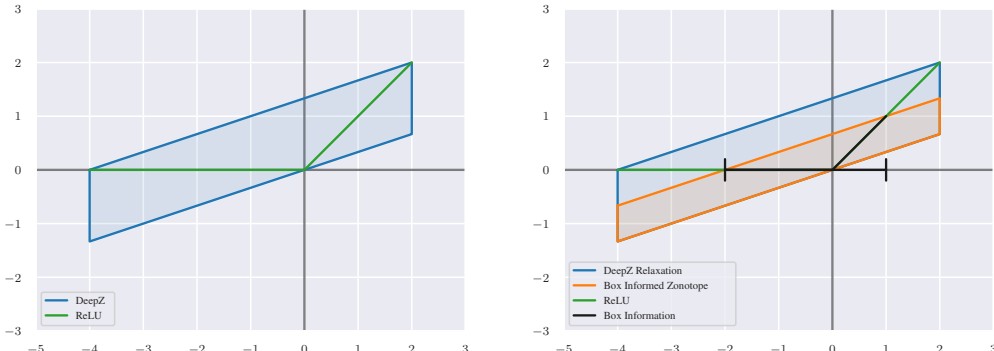

Figure 3: Graphical representation of the coordinatewise bounding procedure of both `DeepZ` and our improved version when box bounds are available. On the left, the blue zonotope is the one introduced by `DeepZ`. On the right, we are also aware than our interval is bounded by the blue-interval, and the blue zonotope is the improved bound.

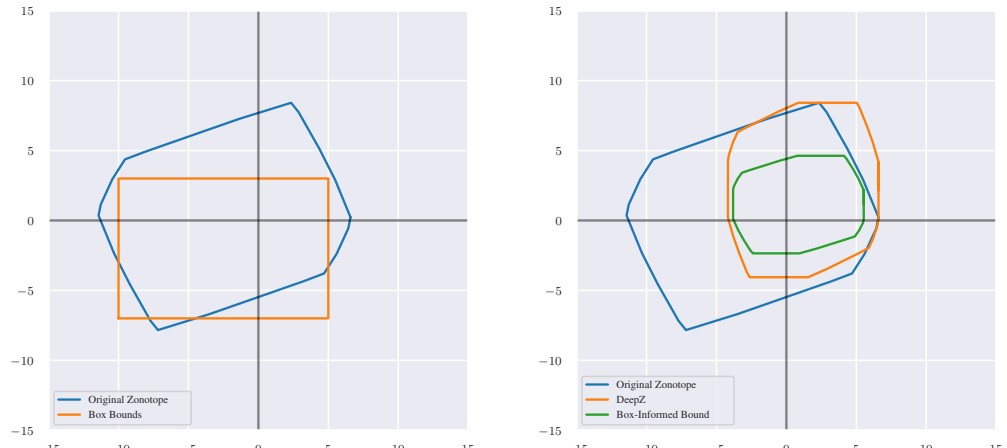

Figure 4: An application of the improved zonotope pushforward operator on a 2-d example. On the left, the original zonotope is outlined in blue, and the extra hyperbox information is drawn in orange. On the right, the output of `DeepZ` is drawn in orange. The output of the improved pushforward operator, that has access to the box, is drawn in green.

this lifting procedure introduces some error. This follows because the graph of the ReLU function, viewed as a 2-d set, is not convex. In this case the canonical triangle relaxation is applied, which is simply the convex hull of the graph of the ReLU function. This triangle relaxation, applied over an interval $[l, u]$ with $l < 0 < u$, has as its upper convex hull, a line with slope $\frac{u}{u-l}$ and intercept $\frac{-ul}{u-l}$. This lifting procedure is iterated over all layers, ultimately yielding a polytope of dimension $dim(\mathcal{X}) + \sum_{i=1}^{L} 2n_i$, where $n_i$ refers to the number of neurons of the $i^{th}$ layer. Optimization over this lifted polytope was framed as a linear program, where the objective was set to minimize the variable corresponding to the final output neuron.

In the original KW formulation, the dual of this linear program, a dual variable was introduced that geometrically corresponds to a lower-bounding linear functional on the triangle relaxation. This is similar to the observation made in $\alpha - \beta - CROWN$ [28]. In the original linear programming paper, an initial choice of $\alpha = \frac{u}{u-l}$ was chosen, which in this case may be interpreted as lower bounding the triangle relaxation by a line that is parallel to the upper convex hull, essentially providing an initial relaxation as a parallelogram with vertical sides.

On the other hand, the zonotope propagation procedure may also be viewed as a polyhedral lifting procedure. Recall in the `DeepZ` formulation, the sound pushforward operator for the ReLU operator works by scaling the $i^{th}$ coordinate by the factor $\frac{u_i}{u_i-l_i}$ and incorporating the error in terms of a

Minkowski sum. This may equivalently be viewed as lifting a $d$-dimensional zonotope, to a $(d+1)$-dimensional zonotope where the newly added dimension has range that is a function of only the $i^{th}$ coordinate. In particular, it is constrained to lie inside the vertical parallelogram defined on the vertical edges by $l, u$, with the slanted edge having slope $\frac{u_i}{u_i - l_i}$. The upper edge is exactly the convex upper hull of the graph of the ReLU function. In practice, because it is convenient and efficient to project zonotopes onto lower dimensional subspaces, when mapping a $d$-dimensional through a ReLU layer, instead of lifting to a $2d$-dimensional zonotope, the original dimensions are simply dropped to retain a $d$-dimensional zonotope.

It is in this sense that the convex outer adversarial polytope described by Kolter and Wong, with the choice of $\alpha = \frac{u}{u-l}$ is exactly equivalent to the zonotope attained by DeepZ if no dimensions were dropped in the zonotope. Hence, each coordinate-wise upper and lower bound attained by each is exactly equivalent. $\qquad\square$

## D    Proof of Hardness of ReLU Programming

**Theorem 4.1.** *When $\mathcal{Z}$ is a zonotope, solving a ReLU program over $\mathcal{Z}$ is equivalent to a neural network verification problem for a 2-layer network, which is known to be NP-hard.*

*Proof.* Let $\mathcal{Z} = Z(c, E)$ be a zonotope in $\mathbb{R}^d$ with $m$ columns in the generator matrix. Then we can construct an equivalent scalar-valued ReLU network, $f(y) = w_2^\top \sigma(W_1 y + b)$. We define $w_2, W_1, b$ as follows:

$$W_1 = \begin{bmatrix} E \\ -E \end{bmatrix} \qquad b = \begin{bmatrix} c \\ -c \end{bmatrix} \qquad w_2 = \begin{bmatrix} c_1 + c_2 \\ -c1 \end{bmatrix}$$

such that $f(y) = c_2^\top \sigma(c + Ey) + c_1^\top (\sigma(c + Ey) - \sigma(-c - Ey))T$, where the ReLU programming objective is recovered by $x = \sigma(x) - \sigma(-x)$. Then we can set the input domain to $[-1, 1]^m$ to equate this verification problem to ReLU programming over $\mathcal{Z}$. The hardness result comes directly from Katz et al [4]. $\qquad\square$

## E    2-Dimensional Zonotopes

Now we turn our attention to 2-dimensional zonotopes, ultimately, we will arrive at a proof of Theorem 4.2 and Corollary 5.1. Throughout, we consider a 2-dimensional zonotope $Z(c, E)$, where $E$ is assumed to have $m$ generators. Further we will assume that the generators are in general position, i.e. two generators are colinear. If this is the case they can be combined into a single generator without changing the set represented by the zonotope. The strategy of this proof will follow in several parts. First we start with a preliminary theorem that holds for zonotopes of all dimension, which describes a relationship between the vertices of the zonotope and the faces of the generating hypercube. Then we show how every two-dimensional zonotope has $2m$ vertices and therefore $2m$ edges. Next we prove that each edge corresponds to exactly one generator, and conversely, every generator corresponds to exactly 2 edges. Finally we describe the procedure we use to enumerate these vertices in a sorted fashion, and compute axis-crossings, origin-containment, and extend this to the intersection of a zonotope and rectangle. We will commonly make use of the notion of the *upper and lower convex hulls* of a 2-dimensional shape. The upper hull of a convex set is the smallest concave function that is larger than the convex hull, and the lower convex hull of a convex set is the largest convex function that is smaller than the convex hull; it is helpful to think of this as the 'upper half' and 'lower half' of the convex hull.

**Lemma E.1.** *Every vertex $v$ of a zonotope with $m$ generators has a corresponding face of the $m$-dimensional $\ell_\infty$ ball that maps to that vertex.*

*Proof.* Consider any zonotope $\mathcal{Z} = Z(c, E)$, which can be equivalently viewed as the affine mapping $y \to c + Ey$ applied to every element in the set $\{y \mid y \in [-1, 1]^m\}$. Consider any vertex $v$ of $\mathcal{Z}$, which by definition must have some vector $a$ (in the polar cone of that vertex), such that $a^\top v < a^\top z$ for all $z \neq v$ in $\mathcal{Z}$. In other words, $v$ is the argmin of some linear program with feasible set $\mathcal{Z}$. Equivalently, the set of $y$ such that $c + Ey = v$ are the argmin of some linear program over the $\ell_\infty$ ball in $\mathbb{R}^m$. This implies that the set $\{y \mid c + Ey = v\}$ is a face of the $\ell_\infty$ ball. Note that face here includes 0-faces such as vertices of the $\ell_\infty$ ball, as well as higher dimensional faces. $\qquad\square$

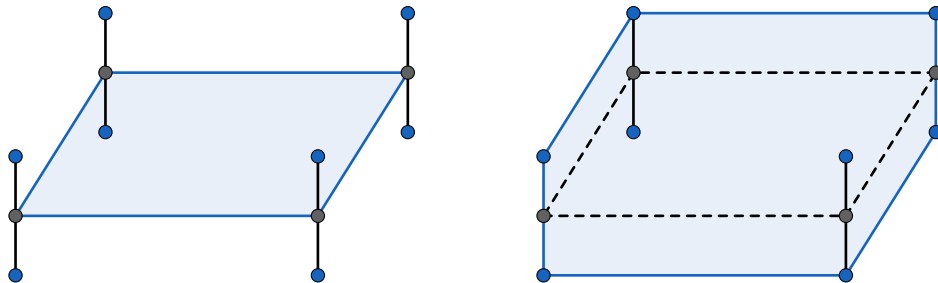

Figure 5: Pictorial aid for Lemma E.2. We consider the following 2-dimensional shapes. The parallelogram on the left is the base case – a zonotope with 2 generators, denoted by the convex hull of the gray vertices only. By incorporating a new vertical generator, the new candidate vertices are colored blue. Taking the convex hull of the blue vertices, we arrive at the 2-dimensional irregular hexagon on the right. Here we see that only the vertices on the left and right endpoints of the upper convex hull yield 2 vertices, whereas the other original vertices only generate 1 new vertex.

**Lemma E.2.** *A 2-dimensional zonotope with $m$ generators in general position has exactly $2m$ vertices and edges.*

*Proof.* We proceed by induction. First note that a zonotope with a two generators is a parallelogram in $2d$ and has 4 vertices and edges. Now assume we have a zonotope $\mathcal{Z}_k$ with $k$ generators, and by the induction hypothesis, $2k$ vertices. Now assume without loss of generality that we add the $k+1^{th}$ generator which is the vector $g := [0, 1]$: we are always able to rotate and scale the zonotope such that this is the case, and invert this operation afterwards. If $V_k = \{v_1, \ldots, v_{2k}\}$ is the set of vertices of $\mathcal{Z}_k$, then $\mathcal{Z}_k$ can be equivalently represented as $\mathrm{conv}(V_k)$. Because adding a generator to a zonotope is the minkowski sum of a segment, $\mathcal{Z}_{k+1}$ can equivalently be written as $\mathrm{conv}(\tilde{V_{k+1}})$, where $\tilde{V_{k+1}}$ is the set of points $\{v_1 + g, v_1 - g, \ldots v_{2k} + g, v_{2k} - g\}$. We now show that only $2k + 2$ of these $4k$ points are vertices. To see this, consider just the vertices of the convex upper hull of $\mathcal{Z}_k$. Because zonotopes are centrally symmetric, there are exactly $k + 1$ of these points. By choosing $g$ to be $[0, 1]$, we see that for every $v_i$ in the convex upper hull, $v_i + g$ is a vertex of $\mathcal{Z}_{k+1}$. We also see that unless $v_i$ is also a point of the convex lower hull of $\mathcal{Z}_k$, $v_i - g$ is not a vertex of $\mathcal{Z}_{k+1}$, because by definition there is an interval vertically beneath $v_i$ such that the entire interval, with $g$ subtracted, is also in $\mathcal{Z}_{k+1}$. Hence for $k - 1$ points of the convex upper hull, we can eliminate $v_i - g$ from the candidate vertex set. We can repeat a similar procedure for the convex lower hull. Hence we are able to eliminate $2k - 2$ points from the $4k$ candidate vertices, leaving exactly $2k + 2$ vertices in $\mathcal{Z}_k$ as desired. Finally notice that every 2-d polytope with $2k + 2$ vertices has exactly $2k + 2$ edges. $\square$

**Lemma E.3.** *If $\mathcal{Z}$ is a 2-dimensional zonotope in general position, then every edge corresponds to exactly one generator. Conversely, every generator corresponds to exactly two edges.*

*Proof.* Pick any edge of $\mathcal{Z}$ and without loss of generality, assume that zonotope is oriented such that the left endpoint of this edge, $v_l$, is the left-most point of the zonotope, neither edge connected to that endpoint is vertical, and the right endpoint of this edge, $v_r$ is above $v_l$. Note that this can always be done and then undone later. Next assume that all generators are oriented such that the $x$-coordinate of each generator is nonnegative. This follows because any generator can be negated without changing the zonotope. From the previous lemma, it can be shown that for zonotope $\mathcal{Z} = Z(c, E)$, the vector $y = -\vec{1}$ satisfies $c + Ey = v_l$. Then, by moving in $y$-space by swapping any coordinate of $y$ from $-1$ to $+1$, we may end up at a new vertex. We can now sort the generators by slope, in terms of $\frac{g(2)}{g(1)}$, where $g(i)$ refers to the $i^{\text{th}}$ coordinate of generator $g$. Suppose $g^{(j)}$ is the generator with the largest slope. Then by interpolating from $y$ to $y + 2e_j$ we traverse along an edge of the zonotope. Indeed, there exists no other direction in $y$-space in which we could move that corresponds to a larger slope-increase in $z$-space. This is essentially a gift-wrapping procedure. Thus, the edge spanned by $v_l, v_r$ corresponds exactly to $g^{(j)}$. To show that every generator corresponds to at least two edges, this follows easily from the fact that zonotopes are centrally symmetric. To show that every generator corresponds to exactly two edges, we can apply Lemma E.2 to show by the pigeonhole principle, no generator can correspond to more than two edges. $\square$

Now with these lemmas in hand, we are prepared to solve the main 2-d vertex enumeration theorem.

**Theorem 4.2.** *If $Z(c, E)$ is a 2-dimensional zonotope with $m$ generators, $Z$ has $2m$ vertices and the set of all vertices, axis crossings, and the containment of the origin can all be computed in $O(m \log m)$ time.*

*Proof.* First we describe how to enumerate all vertices of $Z(c, E)$ in $O(m \log m)$ time. This procedure is a simple extension of Lemma E.3. Recall, in that lemma, we were able to find the left-most vertex by solving a linear program minimizing the $e_1^\top z$ over $Z(c, E)$. Then we were able to find the next vertex, starting from the left and going clockwise, by finding the generator with the largest slope, once generators had been appropriately sign-normalized. We can simply continue this procedure, and from the right-endpoint of the edge enumerated in Lemma E.3, the next clockwise vertex is attained by traversing the generator with the next highest slope. In this way, we can sort the entire set of $m$ generators by their slope in decreasing order and iteratively enumerate the vertices, where the $(i+1)^{th}$ vertex $v_{i+1}$ is simply $v_i + 2g_i$, where $v_i$ is the $i^{th}$ vertex and $g_i$ is the $i^{th}$ generator, in the sorted order. This process generates the $m + 1$ vertices of the entire convex upper hull of $Z(c, E)$ in left-right order. To attain the vertices of the lower convex hull, we rely on the centrally symmetric property of zonotopes: negate each vertex, add $2 \cdot c$, and reverse the order of this list to generate the $m + 1$ vertices of the lower convex hull in right-left order. By concatenating these two lists, and removing duplicates at the endpoints of each convex hull, we are able to generate all $2m$ vertices of the zonotope in clockwise order. The runtime of this algorithm is the time to compute the first vertex and then the slope of each generator, each of which is $O(m)$ time. Then the generator slopes must be sorted in $O(m \log m)$ time. Iterating through this list and adding each generator to the current vertex requires $O(m)$ constant time steps. Computing the lower convex hull only requires $O(m)$ time as well, so the whole procedure requires $O(m \log m)$ time, dominated by the cost of sorting the generators by their slopes.

Now we demonstrate how it takes an additional $O(m)$ time to compute the points on the zonotope where either $x = 0$ or $y = 0$. We consider only the $x = 0$ case, without loss of generality. Since we are given the vertices in clockwise order, it is trivial to walk through the vertex list and report any pair of vertices for which $v_i(1) \cdot v_{i+1}(1) < 0$. If this is the case, then the edge between $v_i$ and $v_{i+1}$ crosses the $y$-axis. Since we have both endpoints on this edge, it requires constant time to compute the point along that line segment for which $x = 0$. A similar procedure can be applied for the $x = 0$ case. Finally, notice that the zonotope only contains the origin if it crosses both the $x$-axis and $y$-axis in two places. $\square$

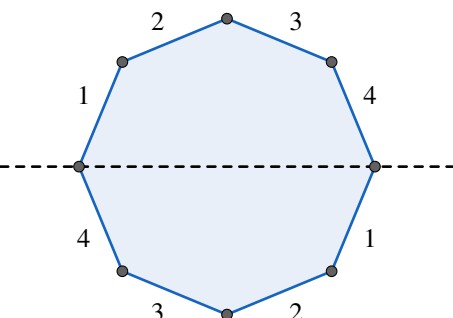

Figure 6: Pictorial aid for Theorem 4.2. By only considering the edges on the upper convex hull (above the dotted line), we sort the edges according to slope and step along them until we reach the end of the upper convex hull. The convex lower hull can be computed easily by central symmetry of zonotopes.

From this procedure, it is now trivial to consider the intersection of a 2-d zonotope and a rectangle.

**Corollary 5.1.** *Given a 2-d zonotope with $m$ generators, the set of candidate optimal points to a ReLU program has cardinality no greater than $(2m + 9)$ and is computable in time $O(m \log m)$.*

*Proof.* First we provide a bound on the cardinality of the set of vertices of the intersection of a 2-d zonotope and a rectangle. The set of candidate vertices is exactly: the vertices of the zonotope, the

vertices of the rectangle, and any intersections of edges of the rectangle and zonotope. Here it is easiest to refer the pictorial aid in Fig. 7. For any rectangle vertex contained in the zonotope, there is necessarily a zonotope vertex not contained in their intersection. For any rectangle vertex not contained in the zonotope, it is possible that up to two new vertices are introduced in the intersection. This means that each vertex of the rectangle can add a net $+1$ to the number of vertices of their intersection. In general, the number of vertices of the intersection of two 2-d polygons is bounded by the sum of the number of vertices of each. Then the candidate optima for a ReLU program over this set is exactly this set of vertices, with cardinality at most $2m + 4$, and any extra axis crossings, and the origin. This brings the set of candidate optima to no more than $2m + 9$.

Next, we describe how to compute this set of candidate optima. We assume that we have obtained the $2m$ vertices of the zonotope in $O(m \log m)$ time. We now check containment of each of the candidate vertices. It requires constant time to compute point-containment within a rectangle, so we can eliminate any zonotope vertices not contained in the box in $O(2m)$ time. It is more complicated to compute point-containment within a zonotope. In general this requires a linear program, but since we already have the vertex list oriented clockwise, we are able to leverage this to compute the set of rectangle vertices contained in the zonotope as well as any edge-edge crossings in $O(m)$ time. The idea is similar to the idea used to compute axis crossings in Theorem 4.2.

The general procedure is to fix an axis and value and shoot a line through that value. For example, we consider the $x = a$ line and compute the intersection of the set $\{(x, y) \mid x = a\}$ with $Z(c, E)$. We can do this by finding the $y$-coordinates of the edges of $Z(c, E)$ that cross this vertical line. This amounts to a single scan through the oriented vertex list of the zonotope, and yields an interval $[l_z, u_z]$ or None. We perform this line-shooting procedure at most 6 times. Once for the left-bound, right-bound, top-bound, and bottom-bound of the rectangle, and if the rectangle itself crosses the $x$ or $y$ axis, we line-shoot the $x = 0$ line or $y = 0$ lines accordingly. For each line-shoot, we are able to compare the interval defined by the box $[l_b, u_b]$ and the line-shoot interval $[l_z, u_z]$. The intersection of this interval can be used to determine which box vertices and edge-edge intersections are contained within the intersection of the zonotope and rectangle. Overall this requires a constant number of calls to a subroutine running in $O(m)$ time, so the overall runtime of this procedure is still dominated by the sort time from the zonotope vertex enumeration, i.e. $O(m \log m)$.

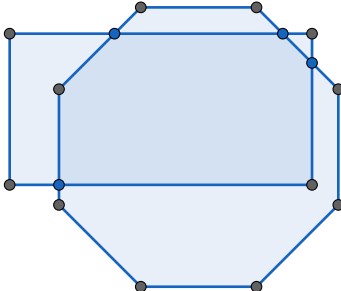

Figure 7: Pictorial aid for the proof of Lemma 5.1. A zonotope (octagon) is intersected with a rectangle. Some rectangle vertices are not contained within the zonotope and can be eliminated, and vice versa. The rectangle vertex on the upper right introduces a net $+1$ vertices to the intersection.

$\square$

## E.1 Partitioning Heuristics

Here we describe the heuristics used to fragment a higher dimensional zonotope into two-dimnesional pieces. We certainly could do this randomly, but we have found some minor performance gains can be achieved by cleverly choosing this partitioning. The key idea for each of these is to choose pairs of dimensions yielding zonotopes that would suffer greatly from being relaxed into a rectangle. Recall that for a two-dimensional zonotope, a rectangle is attained when each generator is colinear with an elementary basis vector. Our two partitioning styles are as follows:

- **Similarity heuristic:** Here we assign each pair of dimensions a scalar-valued score that roughly resembles how far each generator column is from an elementary basis vector. These pairwise scores are encoded in a $d \times d$ matrix. Each row of this matrix is sorted in descending

order, and the partitions are formed by looping over the rows of the matrix and choosing the first available index, that has not already been chosen by a prior pair. The scoring function here, for dimensions $i, j$ is the dot product of the absolute values of the *rows* of the generator matrix. Hence the scoring matrix can be computed by the matrix-matrix multiplication $|E| \cdot |E|^\top$. For zonotopes of very large dimension, this method requires the multiplication of two $(d \times m)$ matrices, and then the sort-and-choose method takes $O(d^2)$ time in the worst case. However since we only have to do this one time, it is often worth it to apply this heuristic on networks with intermediate networks with moderate dimension, or those without spatial information imbued by convolutional layers.

- **Spatial or depthwise heuristic:** For convolutional networks, the structure of convolutional operators can be applied to choose partitionings. In the spatial partitioning heuristic, we assign each coordinate its pair by choosing a coordinate adjacent to it in the feature map. In the depthwise partitioning heuristic, we assign each coordinate its pair by choosing a coordinate in the same location on the feature map but in a different channel. The intuition here is that random partitioning here would often choose coordinates that have disjoint receptive fields and would yield zonotopes that are rectangles. These heuristics much more quickly than the similarity heuristic and are amenable to larger convolutional nets.

# F  Experimental Details

## F.1  Computation Environment

Computations for this paper were performed using the idle resource queue of computing cluster. The cluster uses Intel Xeon Gold 6230R and 6230R CPUs and mixture of NVIDIA 2080 Ti, Quadro RTX 6000, and Telsa A40 GPUs. Due to our usage of the idle resource queue, we ran on a variety of different CPUs and GPUs depending on which resources were being underutilized on the cluster. Our job requirements were 5 CPU cores, 20 GB of RAM, and 1 GPU of any kind per job. Despite our timing information not being representative of a single hardware configuration, we believe we can still make valid comparisons using it, as we ensured that we ran all methods for a particular image on the same node.

## F.2  Network Architectures

### F.2.1  MNIST Networks:

We consider three different architectures trained on the MNIST dataset. Each is randomly initialized, and trained adversarially for 25 epochs using the Adam optimizer and standard hyperparameters to update network weights [34]. The adversarial training was perfromed using 10 iterations of PGD with an $\ell_\infty$ adversarial radius of $\epsilon = 0.1$ [35]. The architectures are as follows, where $\texttt{Conv2D}(in, out, k, s, p)$ refers to a convolutional layer with $in$ input channels, $out$ output channels, a square kernel of size $k$, a stride of $s$, and padding $p$. The Wide and Deep network architecture choices were taken from [11], though the FFNet was our choice for a fully connected network. We trained these networks ourselves.

- $\texttt{MNIST FFNet:}$ (960 ReLU neurons)
    1. $\texttt{Linear}(784, 512) \rightarrow \texttt{ReLU}$
    2. $\texttt{Linear}(512, 256) \rightarrow \texttt{ReLU}$
    3. $\texttt{Linear}(256, 128) \rightarrow \texttt{ReLU}$
    4. $\texttt{Linear}(128, 64) \rightarrow \texttt{ReLU}$
    5. $\texttt{Linear}(64, 10)$
- $\texttt{MNIST Wide:}$ (4804 ReLU neurons)
    1. $\texttt{Conv2D}(1, 16, 4, 2, 1) \rightarrow \texttt{ReLU}$
    2. $\texttt{Conv2D}(16, 32, 4, 2, 1) \rightarrow \texttt{ReLU} \rightarrow \texttt{Flatten}$
    3. $\texttt{Linear}(1568, 100) \rightarrow \texttt{ReLU}$
    4. $\texttt{Linear}(100, 10)$
- $\texttt{MNIST Deep:}$ (5196 ReLU neurons)
    1. $\texttt{Conv2D}(1, 8, 4, 2, 1) \rightarrow \texttt{ReLU}$
    2. $\texttt{Conv2D}(8, 8, 3, 1, 1) \rightarrow \texttt{ReLU}$

3. $\texttt{Conv2D}(8, 8, 3, 1, 1) \rightarrow \texttt{ReLU}$
4. $\texttt{Conv2D}(8, 8, 4, 2, 1) \rightarrow \texttt{ReLU} \rightarrow \texttt{Flatten}$
5. $\texttt{Linear}(392, 100) \rightarrow \texttt{ReLU}$
6. $\texttt{Linear}(100, 10)$

### F.2.2 CIFAR-10 Networks

For the CIFAR-10 networks, we directly utilized the models and weights from prior work [9][†]. Specifically, we considered the models trained both using SGD and adversarially using PGD with the architecture:

- $\texttt{CIFAR WIDE:}$ (6244 ReLU neurons)
    1. $\texttt{Conv2D}(3, 16, 4, 2, 1) \rightarrow \texttt{ReLU}$
    2. $\texttt{Conv2D}(16, 32, 4, 2, 1) \rightarrow \texttt{ReLU} \rightarrow \texttt{Flatten}$
    3. $\texttt{Linear}(2048, 100) \rightarrow \texttt{ReLU}$
    4. $\texttt{Linear}(100, 10)$

### F.3 Hyperparameter Choices

Here we describe the hyperparameter choices for both our model and competing methods. Wherever possible, code from prior works was utilized, with minimal tuning applied. We found the codebase associated with the Active Set paper to be immensely helpful here [9]. Because we changed our partitioning hyperparameters slightly between networks, we describe each hyperparameter choice explicitly. Scripts to run each experiment are contained with the attached codebase. For all experiments, we employ Gurobi as MIP-solver [36].

**Single-stage setting:**

- **BDD+**: For the optimal proximal method, we apply the default parameters where 400 iterations were applied with 2 inner steps and $\eta$ ranging from 1 to 5.

- **AS**: 1000 iterations of AS were increased to 1000 to match our approach, where all other hyperparameters were kept at their default values.

- **ZD (MNIST)**: For the MNIST networks, we initialized with the KW dual variables, used the 'similarity' partitioning heuristic for the 2-d zonotopes, and used 1000 iterations of Adam with default $\beta$ parameters. The learning rate was initialized at $\eta = 0.01$, and decayed by a multiplicative 0.75 factor every 100 iterations.

- **ZD (CIFAR)**: For the CIFAR networks, we initialized with the KW dual variables, used the 'spatial' partitioning heuristic for the 2-d zonotopes, and used 1000 iterations of Adam with default $\beta$ parameters. The learning rate was initialized at $\eta = 0.0025$ and decayed by a multiplicative 0.5 factor every 200 iterations.

- **ZD-MIP**: For all single stage experiments, for every network, we ran the exact same setting as ZD, but merged partitions of the final layer only into zonotopes of dimension 20 and evaluated the dual function. The only exception here is in MNIST FFNET, where only the final layer was merged to zonotopes of dimension 16.

**Stagewise Setting:** In this setting, we kept most hyperparameters the same as in the single-stage setting. The notable exception is a change in the number of iterations in the AS method. We found that 1000 iterations would frequently cause CUDA memory errors, and to the best of our understanding, these methods weren't run in a stagewise setting in [9]. Because these methods are allowed to run for a longer time, we increased the number of iterations dual ascent was performed for, as well as the partitioning at the end. Essentially, these were tuned slightly in order to reliably prove tighter bounds such that AS256-ZD was both faster and tighter than AS512. We summarize the iteration and partitioning changes here :

- **MNIST FFNET**: We initialize as before, using the specified box bounds, and then run 2000 iterations of gradient ascent, with an initial learning rate of 0.0025, decaying by a factor of 0.75 every 400 epochs. Because this method is relatively smaller, we increase the final

---

[†]https://github.com/oval-group/scaling-the-convex-barrier

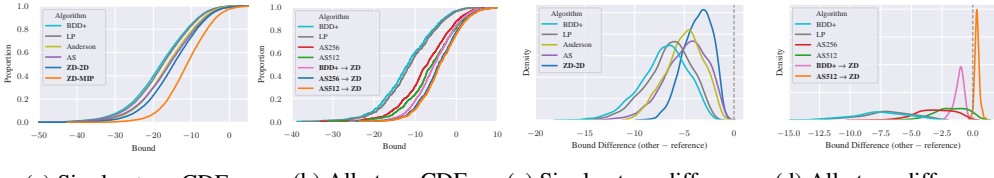

| (a) Single-stage CDF | (b) All-stage CDF | (c) Single-stage difference | (d) All-stage difference |

Figure 8: Full plots for the MNIST FFNet network (zoom in for detail). (a) The CDF plot for the single-stage setting. (b) The CDF plot for the multi-stage setting. (c) The difference plot for the single-stage setting, where the reference method is ZD-MIP. (d) The difference plot for the multi-stage setting, where the reference method is AS256→ZD.

- layer partition size to 32 and run 20 iterations of dual ascent. Finally, we expand all other 2-dimensional partitions to zonotopes of dimensin 16 and evaluate the dual.
- **MNIST WIDE/DEEP**: We initialize and run dual ascent over the 2-d zonotopes as in MNIST FFNet. We then increase partition sizes to 16 for the penultimate layer and 20 for the final layer and evaluate the dual function once.

**Ablation Study Details:** For the ablation study, we only evaluate on the MNIST FFNet. We initialize either with the `DeepZ` bounds, the `DeepZ` bounds augmented via IBP, or the zonotopes augmented by the stagewise BDD+ bounds. Then we partition the zonotopes using the 'similarity' heuristic and run 1000 iterations of gradient ascent with an initial learning rate of 0.01, decaying by a factor of 0.75 every 100 iterations. We increase the partition size to 32 on the final layer only and evaluate the dual. We give results complete with error bars in Table 4.

Table 4: Ablation study for each of phase of ZonoDual. We evaluate the bound and runtime after completion of the Initialization phase (init), Iteration phase (iter), and Evaluation (eval) phases, averaging over the first 1000 examples from MNIST. We report three initialization techniques: DeepZ preactivation bounds, and DeepZ augmented with IBP or BDD+ hyperbox bounds. The computed lower bounds are reported relative to the LP relaxation (a larger number means a tighter bound).

| Init\Phase | Init | Iter | Eval |
| --- | --- | --- | --- |
| DeepZ | $-3.9 \pm 1.1$ | $2.9 \pm 1.0$ | $7.7 \pm 2.6$ |
| Runtime (s) | $0.0047 \pm 0.0004$ | $5.6 \pm 0.3$ | $7.0 \pm 1.5$ |
| DeepZ + IBP | $-3.9 \pm 1.1$ | $2.9 \pm 1.0$ | $7.7 \pm 2.6$ |
| Runtime (s) | $0.024 \pm 0.002$ | $5.6 \pm 0.2$ | $6.9 \pm 1.4$ |
| DeepZ + BDD+ | $5.3 \pm 1.8$ | $6.4 \pm 1.9$ | $10.1 \pm 3.2$ |
| Runtime (s) | $4.9 \pm 0.3$ | $10.5 \pm 0.5$ | $11.5 \pm 1.4$ |

## G  Additional experimental results

Here we have included data from experiments that did not fit in the main paper. Unless otherwise stated, the setup and hyperparameters are chosen as in the main paper and in Appendix F.3.

We present two styles of plots in Figs. 8 to 11. The cumulative distribution plots present the quantity of examples considered that yield a bound less than a specified value. Curves further to the right are better here. The difference plots present the distribution of bound differences relative to a reference method, which is represented by the dotted vertical line. Curves further to the right provide better bounds, and portions of the curve greater than 0 indicate the proportion of examples that the method beats the reference. For example, in the single-stage setting, the curves represent the difference between the technique in the plot and the ZD-MIP method. In the multi-stage setting, we use AS256→ZD as the reference.

We also report numerical timing results in Tables 5 and 6 and bound statistics in Tables 7 and 8.

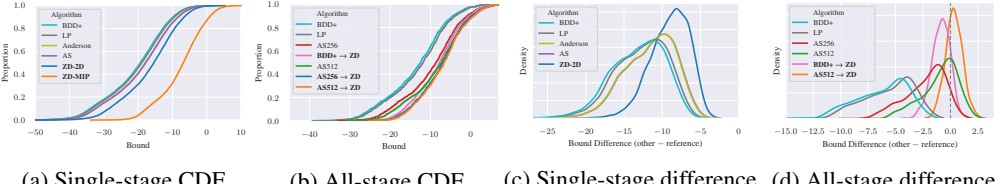

(a) Single-stage CDF    (b) All-stage CDF    (c) Single-stage difference  (d) All-stage difference

Figure 9: Full plots for the MNIST Wide network (zoom in for detail). (a) The CDF plot for the single-stage setting. (b) The CDF plot for the multi-stage setting. (c) The difference plot for the single-stage setting, where the reference method is ZD-MIP. (d) The difference plot for the multi-stage setting, where the reference method is AS256→ZD.

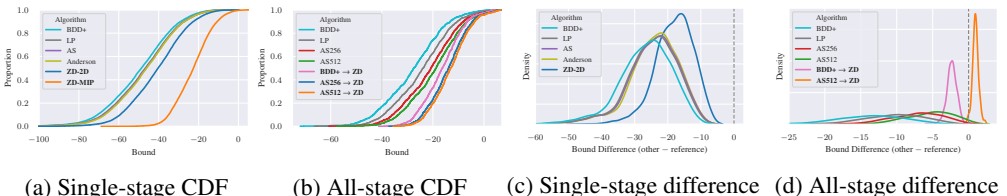

(a) Single-stage CDF    (b) All-stage CDF    (c) Single-stage difference  (d) All-stage difference

Figure 10: Full plots for the MNIST Deep network (zoom in for detail). (a) The CDF plot for the single-stage setting. (b) The CDF plot for the multi-stage setting. (c) The difference plot for the single-stage setting, where the reference method is ZD-MIP. (d) The difference plot for the multi-stage setting, where the reference method is AS256→ZD.

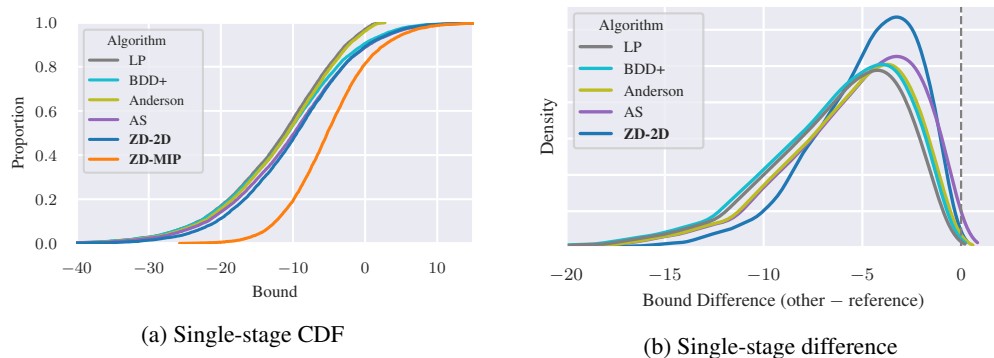

(a) Single-stage CDF               (b) Single-stage difference

Figure 11: Plots for the CIFAR-10 network: (a) CDF plots repeated from the main paper. (b) the difference plots where the reference method is the ZD-MIP method.

Table 5: Time required by each algorithm across a set of four networks in the single-stage setting. Numbers reported are mean and standard deviations in seconds.

| Alg\Net | MNIST FFNet | MNIST Deep | MNIST Wide | CIFAR SGD |
|---|---|---|---|---|
| BDD+ | $1.7 \pm 0.1$ | $2.5 \pm 0.2$ | $1.5 \pm 0.1$ | $1.5 \pm 0.1$ |
| LP | $6.2 \pm 1.1$ | $4.8 \pm 0.5$ | $5.5 \pm 0.5$ | $6.2 \pm 0.6$ |
| **ZD-2D** | $5.7 \pm 0.3$ | $10.9 \pm 0.5$ | $10.2 \pm 0.5$ | $7.6 \pm 0.3$ |
| AS | $10.7 \pm 0.9$ | $16.9 \pm 1.2$ | $10.4 \pm 0.7$ | $10.2 \pm 0.8$ |
| **ZD-MIP** | $6.0 \pm 0.3$ | $23.1 \pm 12.4$ | $14.7 \pm 4.6$ | $11.0 \pm 3.1$ |
| Anderson | $13.1 \pm 2.5$ | $52.4 \pm 27.5$ | $52.4 \pm 19.2$ | $21.3 \pm 4.6$ |

Table 6: Time required by each algorithm across a set of three networks in the all-stage setting. Numbers reported are mean and standard deviations in seconds.

| Alg\Net | MNIST FFNet | MNIST Deep | MNIST Wide |
|---|---|---|---|
| BDD+ | $5.2 \pm 0.5$ | $15.4 \pm 0.4$ | $8.3 \pm 0.3$ |
| **BDD+ $\rightarrow$ ZD** | $52.0 \pm 31.2$ | $51.7 \pm 14.0$ | $36.4 \pm 6.8$ |
| AS256 | $29.5 \pm 0.9$ | $138.9 \pm 0.9$ | $45.4 \pm 0.6$ |
| **AS256 $\rightarrow$ ZD** | $70.0 \pm 27.5$ | $174.1 \pm 13.6$ | $73.3 \pm 6.6$ |
| AS512 | $61.6 \pm 1.8$ | $289.3 \pm 1.7$ | $91.3 \pm 0.9$ |
| **AS512 $\rightarrow$ ZD** | $100.3 \pm 26.6$ | $324.1 \pm 13.5$ | $119.2 \pm 6.9$ |
| LP | $90.0 \pm 16.9$ | $2132.2 \pm 494.4$ | $387.1 \pm 65.5$ |

Table 7: Bounds obtained by each algorithm across a set of four networks in the single-stage setting. Numbers reported are average and standard deviation of the bound relative to **ZD-2D** over the first 1000 images from the respective validation set.

| Alg\Net | MNIST FFNet | MNIST Deep | MNIST Wide | CIFAR SGD |
|---|---|---|---|---|
| BDD+ | $-3.4 \pm 1.1$ | $-8.6 \pm 2.7$ | $-4.2 \pm 1.3$ | $-1.5 \pm 0.9$ |
| LP | $-2.8 \pm 1.0$ | $-6.6 \pm 2.4$ | $-3.7 \pm 1.3$ | $-1.2 \pm 0.9$ |
| Anderson | $-1.3 \pm 0.7$ | $-5.9 \pm 2.3$ | $-2.6 \pm 1.2$ | $-0.7 \pm 0.8$ |
| AS | $-1.2 \pm 0.9$ | $-6.3 \pm 2.4$ | $-2.6 \pm 1.2$ | $-0.5 \pm 0.8$ |
| **ZD-2D** | $0.0 \pm 0.0$ | $0.0 \pm 0.0$ | $0.0 \pm 0.0$ | $0.0 \pm 0.0$ |
| **ZD-MIP** | $3.7 \pm 1.4$ | $18.1 \pm 5.8$ | $9.0 \pm 2.6$ | $4.9 \pm 2.8$ |

Table 8: Bounds obtained by each algorithm across a set of three networks in the all-stage setting. Numbers reported are average and standard deviation of the bound relative to **BDD+ $\rightarrow$ ZD** over the first 1000 images from the MNIST validation set.

| Alg\Net | MNIST FFNet | MNIST Deep | MNIST Wide |
|---|---|---|---|
| BDD+ | $-6.5 \pm 2.1$ | $-11.5 \pm 4.5$ | $-5.4 \pm 2.2$ |
| LP | $-5.9 \pm 2.0$ | $-8.2 \pm 4.0$ | $-4.7 \pm 2.1$ |
| AS256 | $-2.0 \pm 1.5$ | $-5.0 \pm 3.4$ | $-1.4 \pm 1.7$ |
| AS512 | $-0.7 \pm 1.3$ | $-3.1 \pm 3.1$ | $-0.1 \pm 1.5$ |
| **BDD+ $\rightarrow$ ZD** | $0.0 \pm 0.0$ | $0.0 \pm 0.0$ | $0.0 \pm 0.0$ |
| **AS256 $\rightarrow$ ZD** | $1.1 \pm 0.4$ | $2.3 \pm 0.7$ | $0.9 \pm 0.8$ |
| **AS512 $\rightarrow$ ZD** | $1.5 \pm 0.5$ | $3.3 \pm 0.9$ | $1.2 \pm 0.8$ |