# OpenReview forum: "Zonotope Domains for Lagrangian Neural Network Verification"
_NeurIPS.cc/2022/Conference — NeurIPS 2022 Accept_

### Official Review · Reviewer_vLdB · 2022-07-07

**Rating:** 4
**Confidence:** 5
**Soundness:** 2 fair
**Presentation:** 2 fair
**Contribution:** 3 good

**Summary:**

The authors present a novel algorithm to compute bounds on the values of neural network activations.
Bounding algorithms heavily rely on bounds on the network intermediate activations, which are represented as hyper-boxes and neglect interactions across the neurons in the same layer.
Leveraging a modified version of a Lagrangian decomposition from the literature, the authors devise a technique to take zonotope constraints into account in the dual optimization, tightening the resulting bounds yet paying a price in terms of computational efficiency.

**Questions:**

- As commonly done in the related literature, the authors should evaluate their algorithm within BaB to test whether it still pays off for complete verification.

- Run stronger incomplete verification baselines (e.g., alpha-CROWN on both intermediate bounds and output bounds, both via joint and sequential (yet parallel) optimization).

I believe the paper is potentially of great interest to the community, and could greatly benefit from addressing the questions above.

**Limitations:**

Except those associated to the questions outlined above, the authors adequately address the limitations of their work.

**Strengths And Weaknesses:**

**Strengths.**
This is the first paper that employs zonotope domains for intermediate bounds in the context of neural network verification: the idea is definitely novel and interesting, and the authors provide non-trivial solutions to the technical problems associated to the formulation.

**Weaknesses.**
The paper has two major weaknesses: one experimental, the other related to the presentation.
The experiments fail to show that "ZonoDual outperforms the state-of-the-art baselines of the linear programming relaxation and the tightest dual verification methods, in terms of both the tightness of the bounds and runtime.". First of all, the trade-offs of a bounding algorithm need to be assessed within complete verification, which allows to evaluate the tightness of the bounds against their time and memory cost in a fair manner. In Table 4, Beta-CROWN attains significantly tighter bounds in less time than ZD, hinting at disadvantageous speed-accuracy trade-offs.
Furthermore, I am not convinced by the incomplete verification experiments. The authors fail to specify which intermediate bounds are employed for the baselines: I would assume bounds from the Kolter and Wong work, as in [9] and [25]. As shown by the authors, these bounds are necessarily looser than the zonotope domains (yet significantly cheaper to compute), so I believe this is not a fair comparison. Indeed, ZD is typically run for longer than some of the baselines, and zonotope intermediate domains are not guaranteed to be tighter than hyperboxes computed via CROWN or alpha-CROWN (or slope-optimized LiRPA). The authors should take this into account.

Finally, I think the quality of the presentation would be improved if the authors focused less on the supposed dichotomy between primal and dual bounding algorithms (many dual algorithms are formulated from primal relaxations, such as [9] and [25], which they can potentially solve due to strong duality), and more on the fact that they propose to employ zonotope-based intermediate bounds, as opposed to hyper-boxes.


LiRPA: https://arxiv.org/pdf/2002.12920.pdf
Alpha-CROWN: https://openreview.net/pdf?id=nVZtXBI6LNn

---

> ### Author Response · Authors · 2022-08-02
> **Response to Reviewer vLdB**
>
> Thank you for the review and critical questions, though we fear that there may have been some confusion with respect to our contributions. We will address this and offer a more general rebuttal in bullet points below:
> - **“This is the first paper that employs zonotope domains for intermediate bounds in the context of neural network verification”:** To be very clear, we are far from the first work to employ zonotopes as intermediate bounds for neural network verification. We are quite explicit about this in the related work section (see refs [22, 23]). We are, however, the first to employ zonotopic intermediate bounds in the context of dual-based methods for neural network verification: where other approaches use interval/box bounds in intermediate layers.
> - **“The trade-offs of a bounding algorithm need to be assessed within complete verification”:** We argue that there is value in emphasizing the ‘bounding’ phase of branch and bound algorithms specifically. As we have only innovated on the bounding component of verification, we focus our evaluations on this phase, noting that our approach can employ any of the numerous branching strategies and machinery. Moreover, the standard benchmarks for verification use epsilons much smaller than those used for attack/defense papers. This favors techniques with weaker but more efficient bounding algorithms because the depth to which these techniques need to branch is much smaller: indeed, one should think that the branching depth should increase exponentially with the weakness of the bounding algorithm. Hence an important metric to consider is the actual bound attained by a single call of an incomplete verifier.
> - **“The authors fail to specify which intermediate bounds are employed for the baselines”:** In our attempts to be as fair as possible during our evaluation, we have employed the codebase of competing methods or reported numbers directly from prior works. In particular, for the BDD+/LP/AS baselines, we follow the authors of [25] and initialize these methods using the intersection of the Kolter-Wong bounds and IBP. We will make this point explicitly in future revisions.
> - **"These bounds [kolter wong] are necessarily looser than the zonotope domain.”** Yes, but only because zonotopes do not treat each coordinate independently. Proposition 4.1 demonstrates that if we were to replace each of our zonotopes by its smallest-containing hyperbox we would arrive exactly at the bounds attained by Kolter-Wong. The crux of the improvement in our approach is that we are using zonotopes in place of hyperboxes here: we are able to attain the improvements demonstrated in the paper only because zonotopes are able to encode neuron dependencies.
> - **“ZD is typically run for longer than some of the baselines”:** Indeed it is. However most of this overhead in ZD’s runtime comes from evaluating ReLU programs over 2-dimensional zonotopes. This is the price paid for tighter bounds than the faster (but looser) baselines. We note that generating the primal intermediate bounds is extremely fast and completes in a fraction of a second for all of the networks/instances we consider.
> - **“Zonotope intermediate domains are not guaranteed to be tighter than hyperboxes computed via CROWN or alpha-CROWN”:** An important point to consider is that Alpha-CROWN i) implicitly relies upon some form of hyperbox-based intermediate bounds (usually KW or IBP) prior to performing the backsubstitution steps, unless each intermediate bound is optimized either jointly or recursively. Again, incorporating zonotopes into dual-based procedures is our main contribution. And ii) Alpha-CROWN cannot overcome the LP bounds (See figure 3 in the Alpha-CROWN paper), whereas our approach can provide tighter bounds than this so-called “Convex Barrier”
> - **“Many dual algorithms are formulated from primal relaxations”:** Unfortunately there is some overloading of terminology here. By “primal approaches” we specifically mean passing sets through functions in an abstract-interpretation or reachability analysis sense (i.e. hyperboxes, zonotopes, etc). By dual approaches, we mean the methods as in [9] and [25]. This terminology is inspired from Salman et al in the “Convex Barrier” paper (https://arxiv.org/pdf/1902.08722.pdf ).

---

### Official Review · Reviewer_ChY4 · 2022-07-12

**Rating:** 6
**Confidence:** 4
**Soundness:** 3 good
**Presentation:** 3 good
**Contribution:** 3 good

**Summary:**

The paper considers the ReLU-based neural network verification problem, where
it builds on the Lagrangian relaxation of the corresponding optimisation
problem by approximating intermediate layer bounds with zonotopes, which can be
partitioned when evaluating the dual with ReLU constraints. The approach is
experimentally evaluated and compared with the SoA on a number of networks
trained on the MNIST and CIFAR-10 datasets.


**Questions:**

Please see comment on Weaknesses.

**Limitations:**

These are sufficiently addressed.

**Strengths And Weaknesses:**

Strengths

- The method provides balances the strengths and weaknesses of fast but
  imprecise incomplete neural network verification  and exact but slow complete
  verification (for verification purposes this is useful in practice as the
  method can be used after methods with less precision have failed and before
  exact methods are used).

- It so does by connecting Lagrangian decomposition with zonotope-based neural
  network verification in a novel manner.

- The exposition of this is well written and presented.

- Good experimental evaluation.

Weaknesses

- Whilst the efficacy of the method relies on accounting of neuron interactions
  in the dual evaluation, it is not compared with semidefinite programming
  approaches that also account for these interactions.

---

> ### Author Response · Authors · 2022-08-02
> **Response to Reviewer ChY4**
>
> Thank you for raising the point of SDP-based verification methods. As far as we can tell, the SOTA in these techniques is the SDP-FO method of Dathathri et al [https://arxiv.org/abs/2010.11645 ]. We had not explicitly compared our numbers, but the authors of the Beta-CROWN paper had evaluated this method on the same benchmarks we use. Compare Table 3 of [https://arxiv.org/pdf/2103.06624.pdf ] and Table 4 in our submission. Based on these numbers, SDP-FO will likely provide a larger verified robustness than our approach, but will take ~20 hours per example, as opposed to our approach which takes less than a minute.
>
> Along these lines, we had considered using SDP-based approaches to provide efficient and tight solutions to our ReLU programming subproblems (see Definition 4.1), but we found that MIPs yielded similar results and were much more efficient. A breakthrough result for SDP-based verification of 2-layer networks could easily be applied in place of MIPs in our approach.

---

### Official Review · Reviewer_ogn5 · 2022-07-14

**Rating:** 7
**Confidence:** 5
**Soundness:** 3 good
**Presentation:** 4 excellent
**Contribution:** 3 good

**Summary:**

The paper proposes a solution to the problem of neural network verification based on dual Lagrangian optimization. The main insight here is bounding the range of dual variables using Zonotopes to capture inter-neuron dependencies not considered by prior work. However, using Zonotopes for Lagrangian optimization makes the optimization harder. To address this issue, the authors design new algorithms to evaluate the dual function efficiently.

**Questions:**

* In terms of theoretical precision, how does the proposed approach compare with other incomplete verifiers such as PRIMA/beta-crown (when not run in complete mode). Are they incomparable? If yes, then it will be good to investigate other cases where the proposed approach gets better results than these baselines.

* The statement "unlike PRIMA, ZD can directly be plugged into branching frameworks." is not true as the recent work of https://openreview.net/forum?id=l_amHf1oaK uses PRIMA based constraints for branch and bounds.

*  Have you experimented with higher dimensional Zonotopes (e.g., 3 or 4)? or it becomes not feasible beyond the 2D setting? if not, is it possible to maybe compute some sort of approximation (as PRIMA does for k-ReLU) in higher dimension to reduce the cost but still gaining precision?

* In Fig. 5, why not change also the scale factor of the Box informed Zonotope as done in [23]? Does it interfere with the convergence of your method?

* For handling the sigmoid activations, can a MILP approximation (as proposed in https://openreview.net/pdf?id=LQtlOn64EK6Y) be employed to convert the optimization defined over sigmoid functions to ReLU ones?

* "All prior dual approaches have chosen the sets $z_k$ to be axis-aligned hyperboxes" however $\beta$-CROWN is also dual based (as mentioned on line 370) and does not rely on hyperboxes?


**Limitations:**

The authors mention the limitations of their method. The paper does not make any contributions that can have a potential negative societal impact.

**Strengths And Weaknesses:**

Strengths:
+ Neural network verification is a highly relevant problem for the NeurIPS audience. There have been several papers on the topic in recent years at NeurIPS.

+ The writing is rigorous yet understandable. The authors do a great job of presenting their technical ideas at the right level of abstraction.

+ The high-level idea of combining Zonotope analysis with dual Langrangian optimization for NN verification is novel and interesting. However, leveraging this insight to develop a scalable and precise verifier requires tackling several technical challenges. The authors develop solid algorithms to overcome these issues.

Weaknesses:
+ For neural network verifiers, the ultimate metric that matters are the number of verification instances verified. The results in Table 4 show that the method cannot outperform the state-of-the-art. Therefore the usefulness of the proposed approach is unclear.

---

> ### Author Response · Authors · 2022-08-02
> **Response to Reviewer ogn5**
>
> Thank you for the careful review and the very thorough questions. We will address each of them in bullet points below:
> - **“... the ultimate metric that matters are the number of verification instances verified”:** While we generally agree with this statement, bound tightness can often be a more informative metric of interest for several reasons. First, for each example, bound tightness provides strictly more information than a binary indicator variable and can help quantify the improvement of one method over another. More importantly, notice that the standard ‘verification benchmarks’, such as those we have included in Table 4, consider verification tasks for much smaller epsilon (i.e. a much weaker adversary) than is typically considered in the adversarial attack/defense literature. For example, an $\ell\_\infty$ region of 0.026 or 0.015 is much smaller than the threat models usually considered for MNIST. We do not necessarily expect the BaB-based methods to perform as equivalently powerfully on larger threat models. As the difficulty of the task increases, the branching depth must increase, which necessitates an exponential increase in subproblems to be solved. Stronger incomplete verifiers might perform relatively better when the choice of epsilon changes. While we acknowledge our approach is unable to overcome Beta-CROWN on the standard benchmarks, the above reasons justify our emphasis on bound-quality of the incomplete subroutine as opposed to simply reporting verified robustness percentages on a specific choice of epsilon.
> - **“How does the proposed approach compare with other incomplete verifiers such as PRIMA/Beta-CROWN?”.** Beta-CROWN, when run without branching or split constraints, is equivalent to Alpha-CROWN. In this case, this method cannot overcome the convex barrier and should offer performance similar to BDD+ or the LP approach. PRIMA, on the other hand, seems to provide incomparable bounds that are generally empirically tighter, but slower, than our approach.
> - **PRIMA and branch-and-bound:** Thank you for pointing out this reference. We will include this reference and adjust the verbiage in our paper to reflect this new work.
> - **“Higher dimensional zonotopes”:** This reference (https://www.cs.mcgill.ca/~fukuda/760B/handouts/expoly3.pdf) has an upper bound on the number of vertices for a D-dimensional zonotope with M generators being $O(M^{(D-1)})$. Note that this reference does not describe an algorithm to enumerate these even for 2-dimensional zonotopes, which we claim as a contribution in our work. This means that an analogous “closed-form” vertex enumeration solution for ReLU programming would scale superlinearly. In practice, Mixed-Integer Programming is significantly faster, but counterintuitively, due to the overhead of MIPs, it is faster to solve larger-dimensional MIPS than 3 or 4 dimensional MIPS (e.g. 4 10-dim MIPs solve faster than solving 10 4-dim MIPs). Thus we tend to use higher dimensional zonotopes ($d\geq 10$) whenever we employ MIPs.
> - **“Why not change also the scale factor” :** We had considered changing the scale factor as in [23], but noted that it does not seem to make a significant difference in practice versus our approach. Our technique has the added benefit of not generating incomparable bounds so it allows for theoretical claims as in Proposition 4.2.
> - **“For handling the sigmoid activations” :**  While we had focused exclusively on the ReLU nonlinearity, this is a very interesting direction for future work. Yes, we believe that such a sigmoid/tanh MIP approximation could be leveraged during the evaluation phase of our algorithm. More work would be required to argue for efficient evaluation of “Sigmoid Programs” (c.f. Definition 4.1) of 2-D zonotopes during the iteration phase.
> - **“Beta-CROWN is also dual based”:** While Beta-CROWN is dual based, Beta-CROWN also relies (subtly and implicitly) on hyperboxes. Indeed, all “Linear relaxation” methods such as CROWN/Alpha-CROWN/Beta-CROWN have at some basic level a subroutine that requires generating intermediate/pre-ReLU bounds, which imply each pre-ReLU lives in an interval $[l\_i, u\_i]$, for generation of the dual problem. Note that these bounds are treated independently across each neuron, i.e. a hyperbox. We will rephrase this statement in our paper to clarify this point.

---

> > ### Comment · Reviewer_ogn5 · 2022-08-09
> > **Post Response Impression**
> >
> > Dear Authors,
> >
> > Thanks for your response. I agree that there bounds improvement is a relevant metric for showing the benefits of the proposed method, however, I believe that the authors can look into other specifications/networks (beyond the standard ones) where beta-crown, prima etc do not perform as well to show end-to-end benefits. This will make the paper more attractive to other audience not necessarily involved in DNN verification. Given that the authors have addressed my other concerns well, I am happy to increase my gardes.

---

### Official Review · Reviewer_6f9Q · 2022-07-14

**Rating:** 6
**Confidence:** 2
**Soundness:** 1 poor
**Presentation:** 1 poor
**Contribution:** 3 good

**Summary:**

The paper aims at verify outputs of neural networks for a given input range. Based on a branch-and-bound strategy, the paper develops an approach to approximating bounds by solving a verification problem of 2-layer neural networks, which is a subproblem obtained by a branching procedure that performs Lagrange decomposition of an entire verification problem. A key idea of the approach is to use zonotopes, which are expressive and easy to handle computationally.  The experiments demonstrate the usefulness of the proposed approach.  The paper also conducted experiments to demonstrate the usefulness of the approach.

**Questions:**

Can the concerns and questions described as the weaknesses be addressed?

**Limitations:**

Limitations seem to be addressed adequately.

**Strengths And Weaknesses:**

### Strength
- The interesting use of zonotopes for neural network verification.
- The complexity for the computation with zonotopes are provable given.

### Weakness
The presentation is hard to follow.  Specifically, I can't find the following in the paper.
- It is not clear for me how to ensure correctness of the bounds provided by the proposed approach.
- It is not clear how to read the figures and tables in the paper.  Thus, I can't check the claims in the paper is correct.
- It is not clear whether the bounds the paper provides are tighter than those provided by the approach in [25] theoretically or experimentally.  If the latter, where is a formal claim and proof for it?
- Why is Proposition 4.2 important?  In L160, the paper says "The second proposition describes an improved pushforward operator that offers improvements against DeepZ", but I'm not sure why Z' in the proposition is an improvement.

The following is minor.
- What is \sigma in (1c)?
- Why is Wk, not Wk+1, used in (3c)?
- What does "the most negative" mean in L252?

### Post-Response
Some of my concerns were addressed by the author response, so my score raised from 4 (borderline reject) to 6 (weak accept).

---

> ### Author Response · Authors · 2022-08-02
> **Response to Reviewer 6f9Q**
>
> We thoroughly appreciate your review, but are saddened that you have found many components of our paper unclear. We will attempt to offer some clarity in our response and be more careful in our presentation. Please do not hesitate to continue this discussion during the Author-Reviewer Discussion if anything is still unclear.
> - **“How to ensure Correctness”:** We will include a more formal correctness proof in the “Guarantees” paragraph in Section 5 where we discuss our algorithm as a whole. For now, we outline the key ingredients for correctness proof below. By framing verification as a minimization problem, correctness is attained by providing valid lower bounds to the optimization problem. **Fundamentally, we Lagrangify some of the constraints (which correspond to the input/output requirements of each layer) and apply weak duality.** Weak duality always holds and yields a dual function, $g(\rho)$ which attains a form that looks like $g(\rho):= \min_{z \in \cal{Z}} \rho^T h(z)$, where we are hiding many of the details for brevity. Weak duality implies that any choice of dual variables, $\rho$, when fed to $g(\rho)$ lower bounds the original minimization. Intermediate bounds, such as hyperboxes or zonotopes, control the feasible domain $\cal{Z}$ inside the dual function. Tighter intermediate bounds reduce the feasible range of the inner minimization, and therefore increase the evaluation of the dual function. Zonotopes, in general, provide tighter intermediate bounds than hyperboxes, but make the dual function more time-consuming to evaluate. The correctness is ensured so long as we provide a certified lower bound of the dual function, which we do using 2-D zonotopic relaxations.
> - **"How to read the figures and tables in the paper”:** The main quantity we are interested in is the lower bound for the verification optimization problem for each example in the dataset. Given that all algorithms provide a lower bound on the target quantity, the best algorithm is the one that outputs the largest value, i.e., lower bound. There can be many ways to compare different algorithms.
>     +  In Figures 1 and 2 we follow the setup of [25]: for examples with true label i, we consider the difference between the $i^{th}$ and $(i+1)^{th}$ logit of the network (i.e $f(\cdot)\_i-f(\cdot)\_{i+1}$). Larger numbers indicate tighter bounds. Since we can evaluate a bound for each example, we can plot the distribution of bounds as a cumulative distribution function. For example, in Figure 1, each curve indicates the performance of a particular verification algorithm on a large number of examples. A point (x,y) on any curve indicates that that method has a bound tighter than $x$ for $(1-y)\\%$ of the examples considered. Curves further to the right indicate methods that provide tighter bounds. We note that many bounds are negative because we are considering epsilons tighter than those usually considered in the verification baselines. We will clarify this in the revised version of our paper.
>    +  Tables 2 and 3 provide times (in seconds) for each of the considered baselines.
>    + Table 4 considers standard verification benchmarks, where we compare the $i^{th}$ logit against all other logits. We use the standard small epsilon, and report which percent of examples have bounds $\geq 0$ for all logits differences.
> - **“It is not clear whether the bounds the paper provides are tighter than those provided by the approach in [25]”:** Theoretically, we provide a dual optimization whose optimal solution gives a tighter lower bound, but is more time-consuming to find the optimal dual variables. The fundamental difference between our approach and that of [25] lies in the formulation of the dual function $g(\rho)$. Recall that evaluating $g(\rho)$ requires solving an optimization problem over $z\_k’s$. Proposition 4.1 demonstrates that the feasible domain of this optimization problem in our approach (zonotopes) is a subset of the feasible domain of the dual function in [25]. This implies that for any fixed choice of dual variables $rho$, our dual function will provide a tighter lower bound. Empirically, we can not guarantee that we find the optimal choice of dual variable, but we do demonstrate that in aggregate (see figures 1 and 2), the bounds provided by our approach are tighter than those provided by [25].

---

> > ### Author Response · Authors · 2022-08-02
> > **Response to Reviewer 6f9Q  (Continued)**
> >
> > - **“Why is Proposition 4.2 important”:** The primary motivation for employing zonotopes is to provide tighter intermediate primal bounds for use in the dual formulation, as this is one of the key contributing factors to the tightness of the certified lower bound. A primary goal in the paper is to provide tighter intermediate primal bounds for use in the dual formulation. Proposition 4.2 states that in the case where we have incomparable hyperbox bounds in addition to zonotopes bounds (i.e., neither the hyperbox or zonotope is a subset of the other), we can leverage the known hyperbox bounds to attain even tighter zonotopes. This is particularly useful when we generate bounds on each intermediate layer, (“stagewise evaluation”) which provides tighter hyperbox bounds for every layer. Proposition 4.2 allows us to leverage these tighter bounds to attain even tighter zonotopes.
> > - **Minor Questions:**
> >    + Sigma refers to the elementwise ReLU operator. We will include this definition explicitly.
> >    + You are correct: $W\_{k+1}$ should be used in (3c)
> >    + Since we have split up the dual function into many smaller minimization problems, some of them contribute more negatively to the total than others. We are able to choose which subproblem to devote more computational effort to, in order to most efficiently improve the bound.

---

> > > ### Comment · Reviewer_6f9Q · 2022-08-09
> > > **Post Author Response**
> > >
> > > Dear the authors,
> > >
> > > Thank you for the detailed response.  I learned many things from it (an outline of the proof, the experiment results that indicate the practice and usefulness of using zonotopes, the importance of the proposition...) and am leaning towards acceptance. I will raise my score.

---

### Official Review · Reviewer_HtX9 · 2022-07-20

**Rating:** 7
**Confidence:** 4
**Soundness:** 4 excellent
**Presentation:** 3 good
**Contribution:** 3 good

**Summary:**

The paper proposes an approach for neural network verification that combines zonotopes and Lagrangian decomposition. This is done by considering zonotope domains in each subproblem of the Lagrangian decomposition. The paper shows that solving these subproblems under zonotope domains is NP-hard (unlike box domains), and thus proposes a technique to relax the domain to a more tractable set by partitioning coordinates. In particular, it considers 2-dimensional zonotopes which are shown to have good properties in terms of tractability. An extended version merges partitions together, using MIP to solve the problem. Both are shown to produce good bounds at a reasonable time frame compared to other verification approaches.

**Questions:**

Minor details to be addressed:

* The definition of the notation $\sigma$ (the ReLU operator) is missing.
* Please include $z_B$ in (3a) in the minimization terms for clarity.
* In line 284, BDD+ is referred to before it is defined in Section 6; add at least a citation and if possible a sentence explaining what it is (as in Section 6).
* In the initialization phase, it is not clear what the "very minor improvements to DeepZ" are (at least not without looking at the reference). Could you clarify this in the text?
* The main text only specifies the number of neurons for each network (e.g. MNIST-Deep). Given how dependent on depth verification methods are, at least include number of layers, and layer types if possible.
* Typos: "Lagangian" in line 30, "perfrom" in line 326, "bouds" in line 346, "slighctly" in line 359.

**Limitations:**

Appendix B carefully describes the limitations of the work in many ways, which is very much appreciated. I see no other limitations that need to be mentioned.

**Strengths And Weaknesses:**

Neural network verification is a widely studied area and the need for better algorithms is still important, as current relaxations do not scale well to the very large networks used in practice. This paper is a solid step towards that direction, combining two approaches in verification that were previously distinct, zonotope-based bounding and Lagrangian decomposition.

The paper starts off with the simple idea of using zonotope domains in the subproblems of Lagrangian decomposition, and winds up systematically producing interesting results and methods: it proves that these subproblems become NP-hard, finds a more tractable relaxation to solve the subproblems, and shows how to incorporate box bounds in the process. Although these results lean to the incremental side, I consider all of these to be valuable contributions to the area, and as far as I am aware I believe that these contributions are novel. Furthermore, the paper is overall well-written, but there are a few minor details that could be included in the main text here and there (see "Questions" section).

While there is an overhead in considering zonotopes in Lagrangian decomposition, the computational results show that it can push the bound significantly compared to other incomplete verification algorithms. That said, one issue is that it is difficult to know whether this approach is worth it end-to-end as it does not compare with approaches tuned for similar solve times, but it could be viewed as a reasonable middle ground between cheap and expensive methods. Moreover, even if it is not immediately clear as to what situations one might prefer this approach, I can imagine future papers building on this idea as one can consider tighter domains within Lagrangian decomposition.

Overall, while this work is incremental, it executes well in combining the two ideas from the literature, and I do not see any major concerns. Therefore, I recommend acceptance.

---

> ### Author Response · Authors · 2022-08-02
> **Response to Reviewer HtX9**
>
> We thank the reviewer for their careful reading and thoughtful review. We will address each point raised in turn:
> - **“It is difficult to know whether this approach is worth it end-to-end as it does not compare with approaches tuned for similar solve times”**  We generally believe that there exists a Pareto curve within incomplete verification methods: by spending more time, one should be able to attain a tighter bound. While our approach does not occupy either extreme of the Incomplete-verification Pareto frontier, we can still imagine situations where the cheaper-but-looser methods are not sufficient, and the expensive-but-tighter methods are overkill. For example in BaB frameworks, it is not always clear which bounding subroutine will work best: tackling difficult problems with cheap subroutines might require an exponentially large amount of branching. It is our hope that our approach or future works that occupy such a middle ground can be useful for strong verification algorithms.
> - **Minor Details**:
>     + **Typos**:  We will address all typos and clarify the notation. Thank you for your careful reading!
>     + **“Very minor improvements to DeepZ”:** We will clarify this in the text so that this point is self-contained. The improvement is discussed in proposition 4.2. When we have incomparable hyperbox and zonotope bounds (i.e. neither is a subset of the other), we can leverage the extra information of the hyperbox to provide tighter zonotopic bounds in the next layer.
>     + **“Number of neurons for each network”** We outline the full network architectures in Appendix F.2. While it is common in verification literature to simply refer to networks in shorthand by their number of neurons, we agree that depth is an important factor to consider and will make this more clear.

---

> > ### Comment · Reviewer_HtX9 · 2022-08-07
> > **Response to comment**
> >
> > Thank you for the response.
> >
> > Regarding the first point, it is fair to say that there exists a Pareto curve, but I would argue that some of the existing verification algorithms can be adjusted in some way or another to take less or more time (e.g. stopping early), and it would have been very useful to compare against the best verification algorithm that takes roughly the same time. Of course, the caveat here is that if one adjusts a baseline in some way, ensuring that the comparison remains fair is nontrivial. In any case, while I wish this issue were addressed, I do not consider this to be particularly major in light of the other experiments in the paper.
> >
> > I have read the other reviews. I believe that the main weakness is that the method and end-to-end computational result lean to the incremental side, and the end-to-end comparison could have been better developed, but combining the zonotope and Lagrangian duality directions in verification is a valuable contribution nevertheless and in my opinion the execution is solid.
> >
> > I agree with Reviewer vLdB that in the Introduction the sentence "ZonoDual outperforms the state-of-the-art baselines of the linear programming relaxation and the tightest dual verification methods, in terms of both the tightness of the bounds and runtime" is an overstatement. This does not seem to have been addressed yet. This is not only when viewing end-to-end results in Table 4, but also the expression "tightest dual verification methods" is too broad (e.g. exact methods that use duality at some point could fall into this category). I would recommend that the authors make this statement more precise.

---

> > > ### Author Response · Authors · 2022-08-09
> > > **Response to Reviewer HtX9 (cont.)**
> > >
> > > Thank you for continuing the discussion. While fair evaluation is quite complicated and nuanced when controlling for all variables, we certainly see the value in attempting to modify competing methods to control for similar runtimes (possibly by doing some limited branching with faster baselines, or stopping early with slower baselines). We can add experiments to this effect in future revisions.
> > >
> > > Regarding the precision in our claims: we will adjust these sentences to more accurately reflect our results. A more precise claim would be to say that "ZonoDual outperforms the linear programming relaxation in both tightness and runtime, and yields a tighter bounding algorithm than the prior dual approaches of [25, 9] (BDD+, AS)." In general, we will clarify that "tightest dual verification algorithms" refers to the bounding approaches of dual-based verifiers in [25, 9], and not their complete/branching variants.

---

### Meta-Review · Area_Chair_ErYB · 2022-08-26

**Recommendation:** Accept
**Confidence:** Certain

**Metareview:**

The authors develop a novel approach for verifying input-output properties of neural networks like adversarial robustness, combining the advantages of earlier work on abstract interpretation and Lagrangian decomposition.

The reviewers agree that the paper presents a novel, sound and significant contribution to the literature on NN verification. One reviewer had concerns regarding whether the approach actually shows substantial benefits when used as part of a complete verifier in terms of the number of verified instances, but agreed that this is likely problem dependent and does not take away the significance of the work. Another reviewer raised more serious concerns that the author should address in the final version of the paper. In particular:
1) Fairness of comparing against baselines with weaker intermediate bounds: The authors should revise the paper stating that the baselines they compare against depend on the quality of the intermediate bounds and the SOTA versions of these methods use tighter intermediate bounds than the ones used by the authors.
2) Highlight the issues with tightness of verification vs speed of verification tradeoffs in the context of complete verification (ie branch and bound), and discuss how the authors' approach might compare with approaches like alpha-beta CROWN.

**Award:**

No

---

### Decision · Program_Chairs · 2022-09-14

Accept